# A neuromorphic physiological signal processing system based on VO$_2$ memristor for next-generation human-machine interface

Rui Yuan[1,6], Pek Jun Tiw[1,6], Lei Cai[1], Zhiyu Yang[2], Chang Liu[1], Teng Zhang[1], Chen Ge [3], Ru Huang [1] & Yuchao Yang [1,2,4,5] ✉

Physiological signal processing plays a key role in next-generation human-machine interfaces as physiological signals provide rich cognition- and health-related information. However, the explosion of physiological signal data presents challenges for traditional systems. Here, we propose a highly efficient neuromorphic physiological signal processing system based on VO$_2$ memristors. The volatile and positive/negative symmetric threshold switching characteristics of VO$_2$ memristors are leveraged to construct a sparse-spiking yet high-fidelity asynchronous spike encoder for physiological signals. Besides, the dynamical behavior of VO$_2$ memristors is utilized in compact Leaky Integrate and Fire (LIF) and Adaptive-LIF (ALIF) neurons, which are incorporated into a decision-making Long short-term memory Spiking Neural Network. The system demonstrates superior computing capabilities, needing only small-sized LSNNs to attain high accuracies of 95.83% and 99.79% in arrhythmia classification and epileptic seizure detection, respectively. This work highlights the potential of memristors in constructing efficient neuromorphic physiological signal processing systems and promoting next-generation human-machine interfaces.

Physiological signals reflect the electrical activity of a specific body part[1] and provide valuable information about mood, cognition, and many other health issues[2], thus any deviation from the norm in patterns may indicate an underlying health problem. For instance, arrhythmias can be picked up by electrocardiogram (ECG) signals[3] while epilepsy, which is a common neurological disorder, manifests itself as abnormalities in electroencephalogram (EEG) signals during epileptic seizure[4]. Monitoring and analyzing these physiological signals form the basis of biomedical devices used for the diagnosis, detection, and treatment of various diseases[2]. While anomaly detection and analysis can be done manually, a physiological signal processing system that is capable of providing diagnosis without human intervention can be useful in providing a second opinion or even picking up subtle and easily overlooked patterns.

In a traditional physiological signal processing system, the analog physiological signals are first converted into digital signals by analog-to-digital converters (ADC) and then stored in memory before being further processed in digital computing units[5–7]. However, the frequent

[1]Beijing Advanced Innovation Center for Integrated Circuits, School of Integrated Circuits, Peking University, Beijing 100871, China. [2]School of Electronic and Computer Engineering, Peking University, Shenzhen 518055, China. [3]Beijing National Laboratory for Condensed Matter Physics, Institute of Physics, Chinese Academy of Sciences, Beijing 100190, China. [4]Center for Brain Inspired Chips, Institute for Artificial Intelligence, Frontiers Science Center for Nano-optoelectronics, Peking University, Beijing 100871, China. [5]Center for Brain Inspired Intelligence, Chinese Institute for Brain Research (CIBR), Beijing, Beijing 102206, China. [6]These authors contributed equally: Rui Yuan, Pek Jun Tiw. ✉e-mail: yuchaoyang@pku.edu.cn

movement of a massive amount of data between memory and computing units heavily affects the speed and power consumption[8,9]. The parallel and event-driven[10] neuromorphic computing system, which is inspired by the human brain, is a promising alternative approach for breaking the von Neumann bottleneck[11]. It is much more energy efficient and suited for processing physiological signals, as they contain spatiotemporal information, thus motivating the design of a brain-like physiological signal processing system. Although some neuromorphic physiological signal processing systems based on complementary metal-oxide semiconductor (CMOS) technology have been demonstrated, most of them suffered from area and energy inefficiencies, due to the incorporation of complex auxiliary circuits and bulky capacitors for the implementation of bio-dynamics[11–15].

To achieve an efficient neuromorphic physiological signal processing system, memristors provide an appealing platform due to their abundant ion dynamics[16–28] and electrical behaviors akin to those found in biological neurons and synapses, hence lending themselves well to realizing compact neuromorphic architectures. While the hardware implementations of Leaky Integrate and Fire (LIF) neurons have been reported widely in literature[27–29], few studies demonstrated hardware implementations of Adaptive-Leaky Integrate and Fire (ALIF) neurons[30,31], which have been shown to improve the computational capabilities of neuromorphic systems. Nevertheless, these ALIF hardware implementations still have room for optimization. More importantly, a complete memristor-based neuromorphic physiological signal processing system that features a highly efficient spike encode scheme and a more biologically plausible neural network with ALIF neurons has not yet been reported.

In this work, a complete neuromorphic physiological signal processing hardware system for the next-generation human-machine interface based on $VO_2$ memristors is demonstrated. Specifically, a platform that can convert analog physiological signals into a stream of asynchronous spike events is proposed, which fully utilizes the positive and negative symmetric thresholds and fast volatile characteristics of $VO_2$ memristors so as to simplify the circuit. Different from the frequency-encoding mode of traditional neurons, the spikes from each channel of the encoding platform mark the time at which the input signal has changed beyond a fixed threshold, which can preserve the original input information content to the greatest extent while keeping a low spiking rate to reduce energy consumption. Besides, a memristor-based decision system that features a Long short-term memory Spiking Neural Network (LSNN) with powerful computational capabilities[32] is provided, wherein ALIF neurons were designed efficiently using $VO_2$ memristors. The performance of this system was evaluated via arrhythmia classification and epileptic seizure detection tasks, achieving accuracies of 95.83% and 99.79%, respectively. This system with a small LSNN has implied immense potential in processing various physiological signals and can hold great prospect in dealing with other temporal signals in general.

## Results

### Design of $VO_2$ memristor-based neuromorphic physiological signal processing system

Figure 1 schematically illustrates the proposed $VO_2$ memristor-based neuromorphic processing system, which integrates an asynchronous spike encoder and an LSNN-based decision system. In the data compression and encoding stage, the memristor-based asynchronous spike encoder converts each channel of the collected physiological signals, such as ECG and EEG, into two-channel spike trains (UP/DOWN channel), which represent the rise or fall of the original signal, respectively. The asynchronous spike encoder based on memristor was inspired by LC-ADC[14,33–36] and delta modulator circuits[37–39], wherein spikes from each channel mark the time at which the input signal changes beyond a fixed threshold. The speed of spikes emission is

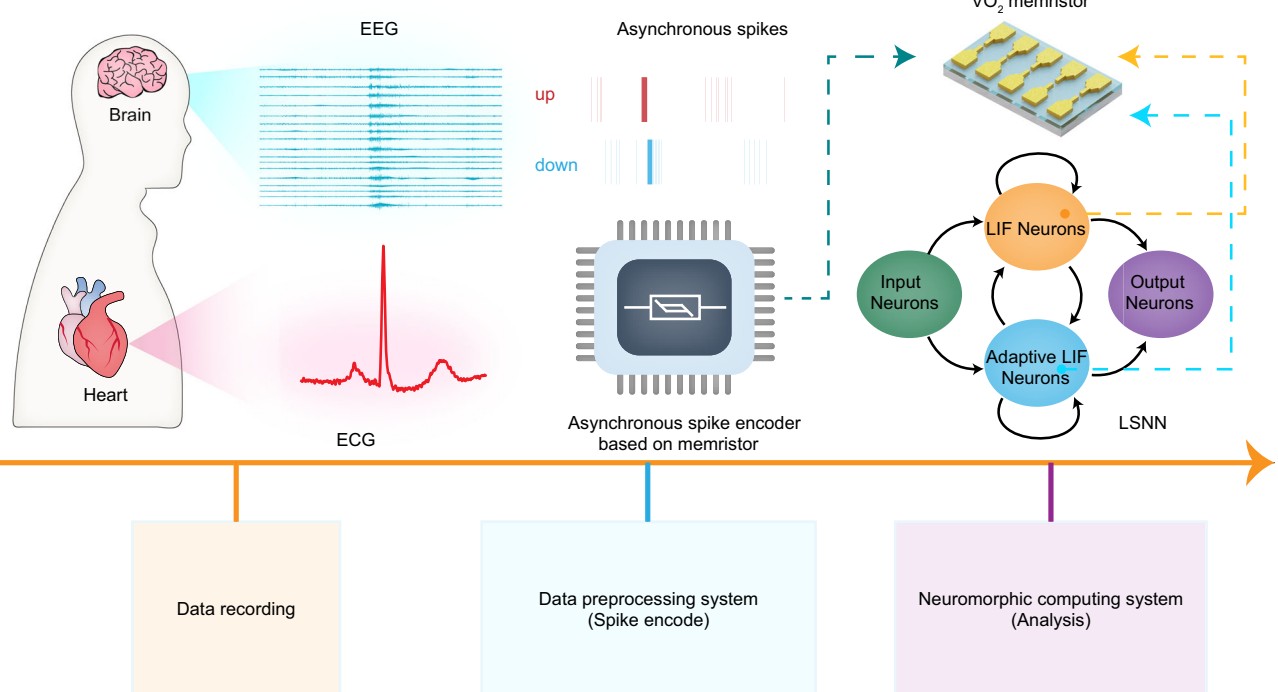

**Fig. 1 | The neuromorphic physiological signal processing system based on $VO_2$ memristor for the next-generation human-machine interface.** The system integrates an asynchronous spike encoder and a decision system based on memristors. In the data compression and encoding stage, the memristor-based asynchronous spike encoder converts each channel of the recorded physiological signals, into two-channel spike trains (UP/DOWN channel). Spikes from each channel mark the time at which the input signal changes beyond a fixed threshold, thus realizing non-uniform and sparse spike coding to reduce the amount of data and energy consumption. In the processing stage, a decision system featuring an LSNN is used to process encoded spike information and to output results, wherein the LIF and ALIF neurons are also constructed by $VO_2$ memristors efficiently.

determined by the variation rate of the input signal, thus realizing non-uniform and sparse spike encoding, which can reduce the amount of data and energy consumption. Compared with the frequency encoding of traditional neurons, this method contains temporal information and is thus more friendly to neuromorphic systems. Since the information of the original signal is preserved to the greatest extent, the encoded asynchronous spike trains can reconstruct the original signal accurately, which is hard for frequency coding. In the section regarding the asynchronous spike encoder based on VO2 memristor, we introduced in detail how to use memristors to implement asynchronous spike encoding without ADC/DAC and special control circuits. Another core of the system is the decision network. Here, a memristor-based decision system that features an LSNN is utilized in which the memristor also plays a central role. The LSNN-based decision system contains two kinds of neurons, the LIF neuron and ALIF neuron. Among them, the hardware implementation of the ALIF neuron requires a feedback mechanism and hence is relatively difficult. In this system, we utilized VO2 memristors to construct not only LIF neurons but also ALIF neurons efficiently. By incorporating these key characteristics, the neuromorphic system can exhibit high accuracy with few weights and used for physiological signal processing in human-machine interfaces.

## VO2 memristor-based artificial LIF neuron

Neurons are the building blocks of brain-like systems. To construct the artificial neuron efficiently, memristors with highly uniform threshold switching (TS)[27,40,41] and volatile characteristics are required. The memristor used in this work is based on VO2 and is designed as a planar device as shown in Fig. 2a. Supplementary Fig. 1a shows a scanning electron microscopy (SEM) image of the device, where the channel length is 400 nm and the electrode width is 2 µm. Details of fabrication processes are shown in Methods. Supplementary Fig. 1b shows the transmission electron microscopy (TEM) image of the device, and a zoom-in view of the VO2 film and corresponding fast Fourier transformation is shown in Supplementary Fig. 1c, d, where well-ordered lattice fringes are evident, verifying the high crystalline quality of VO2 film which is important for achieving high uniformity in our devices. The cross-sectional scanning transmission electron microscopy (STEM) image and corresponding energy dispersive X-ray spectroscopy (EDS) mapping of O, Al, Si, V, Ti, and Au elements in the device can be seen in Supplementary Fig. 2a, along with EDS elemental line profile in the same region (Supplementary Fig. 2b). Stable volatile resistive switching is indicated by the I–V characteristics of the VO2 memristor (Fig. 2b), where 100 cycles were performed. The device changes from a high resistance state (HRS) to a low resistance state (LRS) once the applied voltage exceeds a threshold voltage ($V_{th}$) of around ±3.4 V and immediately returns to HRS when the applied voltage falls below the holding voltage ($V_{hold}$) of around ±1.45 V. This resistive switching phenomenon arises from the metal-insulator transition of VO2, which is a result of the intertwined structural and electronic phase changes[42–44]. The transition between the low-temperature semiconducting phase and the high-temperature metallic phase occurs at around ~340 K, and can be triggered by Joule heating[45]. To illustrate this point, we simulated the thermodynamic resistive switching process using COMSOL Multiphysics. As shown in Supplementary Fig. 3, the switching of the VO2 memristor between HRS and LRS is accompanied by the formation or disappearance of a high-temperature filament, which has also been previously observed[46,47]. To

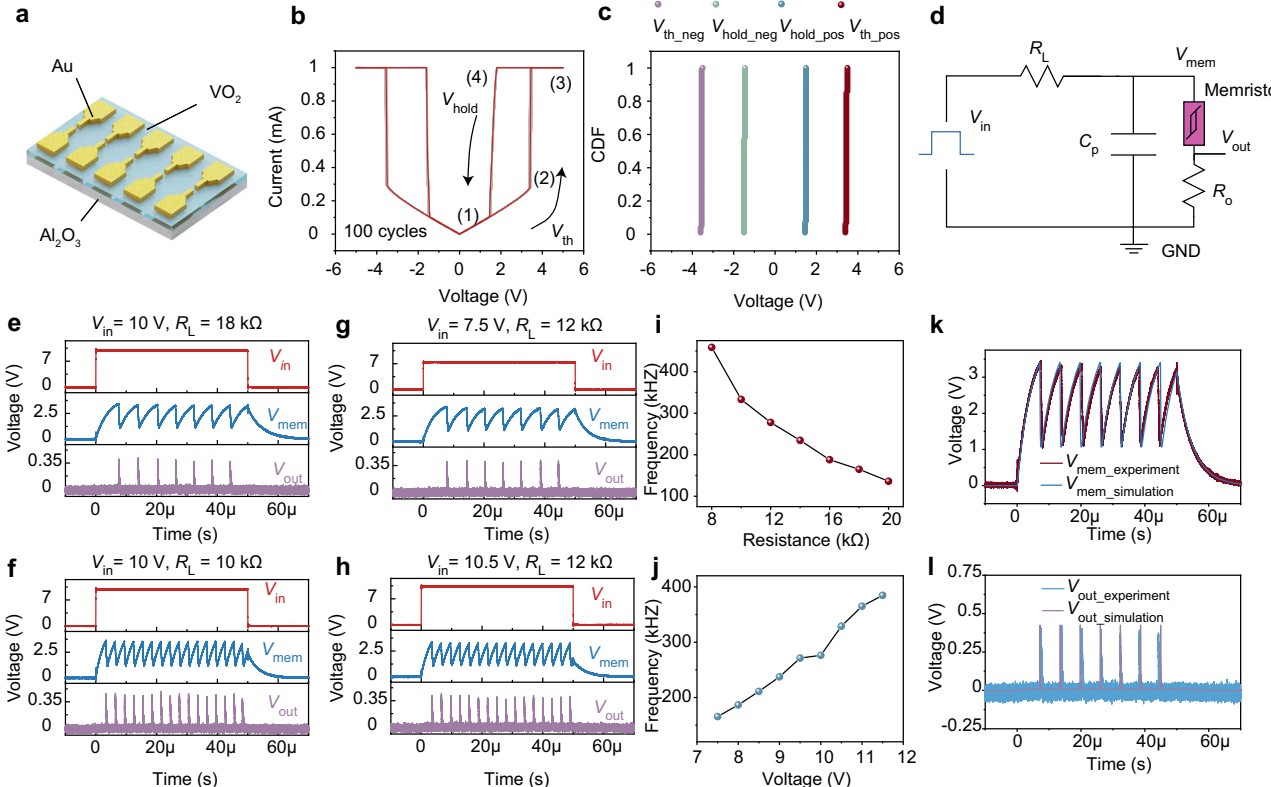

**Fig. 2 | The implementation of memristor-based artificial LIF neuron.**
**a** Schematic diagram of the VO2 memristor, which is designed as a planar structure. **b** Current-voltage characteristics of the device repeated for 100 cycles, showing stable volatile resistive switching. **c** Cumulative plots of $V_{th\_pos}$, $V_{hold\_pos}$, $V_{th\_neg}$, and $V_{hold\_neg}$, showing low variations. **d** Illustration of the artificial neuron based on VO2 memristor. **e**–**h** The artificial neuron response under different series resistance $R_L$ and applied voltage. **i, j** The effect of series resistance $R_L$ and applied voltage on spiking frequency. Larger series resistance and lower input voltage will result in a lower charging current, thereby reducing the firing frequency. **k, l** Simulation results of the artificial neuron using our SPICE model, showing high congruence with experiment results.

be specific, heat is generated in the $VO_2$ memristor as the applied voltage increases (state (1) to (2)). Once the phase transition is triggered, a filament forms through the $VO_2$ gap, switching the device from HRS to LRS. Then, the filament expands as the voltage is increased (state (2) to state (3)). When the voltage is reduced, the heat dissipates, and the filament size decreases (state (3) to state (4)). Once the applied voltage is below $V_{hold}$, the filament breaks down and the device eventually returns to HRS (state (4) to state (1)). The simulated $I$-$V$ curve agrees well with the experimentally measured curve, further verifying the Joule heating-induced phase transition and the filament formation picture. Figure 2c displays the cumulative plots of positive and negative threshold/holding voltages, including $V_{th\_pos}$, $V_{hold\_pos}$, $V_{th\_neg}$, and $V_{hold\_neg}$ in 100 repeated cycles. The coefficient of variation ($C_v$) defined by the ratio of the standard deviation ($\sigma$) to the mean value ($\mu$) of $V_{th\_pos}$, $V_{hold\_pos}$, $V_{th\_neg}$, and $V_{hold\_neg}$ were 0.65%, 0.86%, 0.31%, and 1.68%, respectively, showing very low cycle-to-cycle (C2C) variations. The superior uniformity can be attributed to the high crystallinity epitaxial $VO_2$ thanks to the matching lattice planes across the film-substrate interface[48], as well as the preservation of such desirable qualities in a planar device structure (Supplementary Note 1). In addition to the uniformity observed under steady state, the $VO_2$ memristor also displayed very small variations in $V_{th}$ and $V_{hold}$ when it was connected to an external circuit and was operating in a dynamical state (Supplementary Fig. 4). The $C_v$ of $V_{th}$ and $V_{hold}$ during ~1000 periods of transient oscillations were 0.73% and 0.48%, respectively. Moreover, when the planar $VO_2$ memristor was operated in air under normal atmospheric pressure, under different ambient pressures ranging from $3.5 \times 10^{-3}$ mbar down to $5.0 \times 10^{-4}$ mbar and in an $N_2$ environment, it also exhibited stable threshold switching behavior with no appreciable difference in its $I$-$V$ characteristics (Supplementary Fig. 5). This implies that such devices are not affected by various atmospheric content such as moisture. Crucially, the $VO_2$ memristor demonstrated a high endurance of $>6.5 \times 10^6$ switching cycles (Supplementary Fig. 6), which ensures the reliability of encoders and neurons that incorporate such devices. Supplementary Fig. 7 displays the transient electrical measurements, where the switching speed of the $VO_2$ memristor in this work is <70 ns from off-state to on-state and <60 ns from on-state to off-state, exhibiting a high-speed threshold switching characteristic.

The circuit configuration of artificial neuron based on $VO_2$ memristor is shown in Fig. 2d. The $VO_2$ memristor is connected in parallel with a capacitor and in series with a load resistor $R_L$. Besides, a 50 Ω resistor $R_0$ is used to convert the current into a voltage output. The dynamics of an ion channel located near the soma of a neuron can be mimicked by the threshold switching (TS) behavior of $VO_2$ while the membrane capacitance is represented by $C_p$. The oscilloscope is used to measure electrical waveforms across the $C_p$, the input waveforms, and the output of the artificial neuron (see Methods and Supplementary Fig. 8). When a voltage is applied to the artificial neuron, the capacitor begins to charge. Once the voltage on the capacitor exceeds $V_{th}$, the $VO_2$ memristor switches to LRS. As a result, a spike is generated, which will be transmitted to the next neuron. Besides, the capacitor will be discharged through the on-state memristor. When the voltage on the capacitor drops below $V_{hold}$, the device will return to HRS. The spiking rate of the artificial neuron strongly depends on the series resistance, applied voltage, and parallel capacitance. Figure 2e, f shows the response of the artificial neuron under different series resistance $R_L$ (18 kΩ, 10 kΩ) when fixing a constant input voltage of 10 V without an external parallel capacitor (More results can be found in Supplementary Fig. 9). A larger $R_L$ will reduce the input current, thus slowing down the charging process, thereby reducing the firing frequency (Fig. 2i). On the other hand, a larger input voltage will increase the charging current, thereby speeding up the charging process, thus increasing the frequency (Fig. 2g, h, j and Supplementary Fig. 10). Supplementary Fig. 11 shows the experimental response of the artificial

neuron under different parallel capacitors. As the parallel capacitance increases, the integration process becomes slower, thus reducing the firing frequency. These firing behaviors can also be deduced from the RC circuit analysis detailed in Supplementary Note 2.

To gain insights into the neuron circuit behavior and to assist in designing the ALIF neuron and the spike encoder, we developed a SPICE model using LTSPICE for our $VO_2$ memristor (Supplementary Fig. 12 and "Methods") based on the one proposed in ref. 49. Our improved model has no polarity, thus allowing symmetrical static $I$-$V$ characteristics and switching thresholds under positive and negative biases, which is in accordance with practical planar $VO_2$ memristors. In essence, the model consists of a comparator, which compares the terminal voltage of the device to $V_{th}/V_{hold}$ when it is in HRS/LRS and flips the state if the terminal voltage increases/decreases beyond the thresholds. The inclusion of $R_0$ and $C_0$ is to suppress instantaneous state transitions of the comparator, which models the finite switching time of the real-world $VO_2$ memristor. The voltage across $C_0$ is then used to determine the resistance of the $VO_2$ memristor model. The simulation results were in good agreement with the experimental results as shown in Fig. 2k-l, where $V_{in}$ and $R_L$ were set as 10 V and 18 kΩ, respectively (Supplementary Table 1 lists the parameters of the device).

## $VO_2$ memristor-based adaptive LIF neuron

Building upon the compact LIF neuron presented in the previous section, we designed a highly efficient $VO_2$ memristor-based ALIF neuron by adding an adaptive control circuit, which requires only a few extra components and a feedback connection (Fig. 3a). The adaptive property stems from the increased membrane leakage current after the neuron fires, which renders subsequent input integration harder and has its analogous process found in biological neurons[50]. The key processes involved in this ALIF neuron are summarized in Fig. 3b. The workings of the LIF part are similar to that of the LIF neuron, but with an additional membrane leakage path via $M_3$. To achieve adaptation, the spike output is amplified by the $M_1$ common-emitter amplifier to drive $M_2$, which charges $C_2$ and increases $V_g$. Consequently, $M_3$ turns on more and increases the leakage current with each spike. By Kirchoff's Current Law, the increased leakage current is subtracted from the input charging current, resulting in a slower charging of $C_1$ during the integration phase and a reduced spiking frequency. In biological neurons, the adaptive effect diminishes and the spiking frequency returns to the initial level when the neuron is rested. In our ALIF neuron, this feature is enabled by $R_3$, which provides $C_2$ with a discharging path and turns off $M_3$ slowly. It is important to note that $V_g$ has to be a slow-changing variable relative to $V_m$, that is to say, the adaptive time constant ($\tau_a = R_3 C_2$) needs to be sufficiently large compared to the membrane time constant ($\tau_m = (R_{VO2} + R_1)C_1$). To effectively utilize the temporal processing capability of ALIF neurons, which stems from their adaptive property, the choice of $\tau_a$ should roughly be on the same time scale as the total input duration[31,51]. Another point to note is that although not demonstrated in this work, the simple common-emitter amplifier introduces signal gain and allows the neuron to drive subsequent stages[23], which could be beneficial in realizing future compact multilayer networks.

To further understand the workings of this ALIF neuron, we simulated the circuit in LTSPICE with the aforementioned $VO_2$ model. The circuit parameters used under controlled conditions are listed in Supplementary Table 2. To illustrate the effect of $V_g$ on the spiking frequency, we directly varied the voltage applied on the gate of $M_3$, and calculated the spiking frequency and its reciprocal, the inter-spike interval ($ISI$), from the resulting output voltage spikes (Fig. 3c). When $V_g$ is lower than the turn-on threshold voltage of $M_3$ ($V_{t, M3}$ ~ 0.7 V), $M_3$ is off and the spiking frequency remains constant in this range. As $V_g$ is increased further beyond $V_{t, M3}$, the spiking frequency decreases monotonically. Beyond a $V_g$ of ~1.65 V, the membrane leakage current is

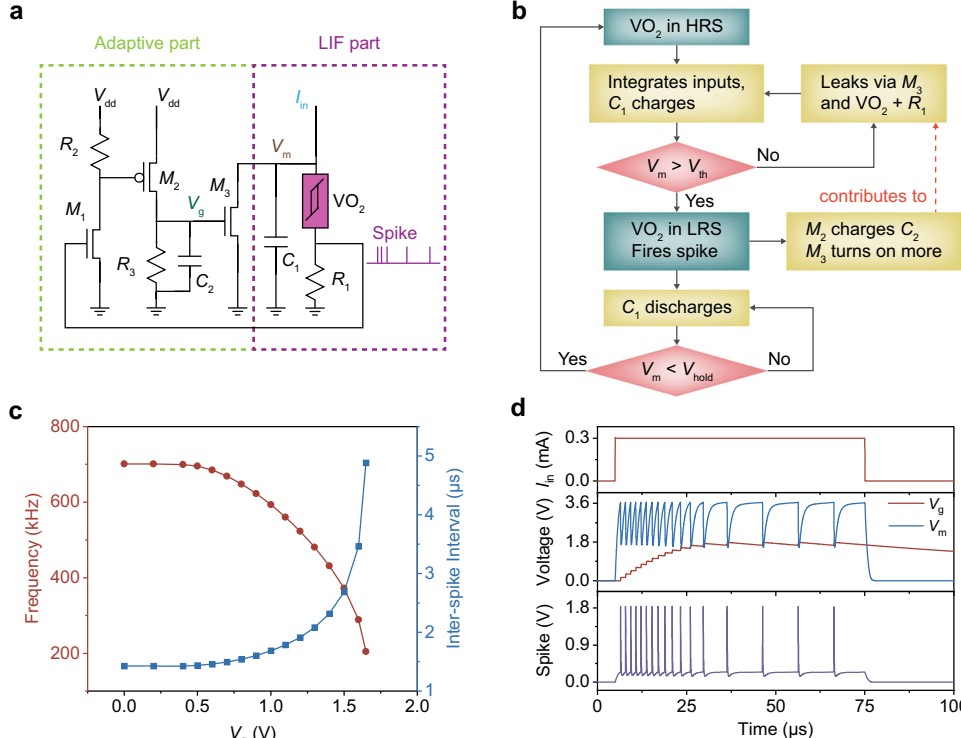

**Fig. 3 | Design and characteristics of VO₂ memristor-based adaptive LIF neuron.**
**a** Circuit schematic of the VO₂ memristor-based ALIF neuron. **b** Flow chart of the operation of the ALIF neuron. Each spike from the LIF part results in the charging of $C_2$ and an increase in $V_g$, which turns on $M_3$ more, contributing to an increased leakage current, and thus adaptation. **c** Effect of $V_g$ on the spiking frequency and its reciprocal, the inter-spike interval (*ISI*), of the neuron. At small $V_g$ (<$V_{t, M3}$ at -0.7 V), $M_3$ is off and the frequency is independent of $V_g$. As $V_g$ is increased further, the

frequency decreases monotonically. Beyond a $V_g$ of -1.65 V, $V_m$ cannot reach $V_{th}$ and the neuron stops firing. **d** Adaptive response of $V_g$, $V_m$, and $V_{spike}$ when a 0.3 mA input current is applied, showing the gradual increase of $V_g$. The frequency remains constantly high at first, before decreasing and finally saturating at a low level. When the adapted frequency saturates, $V_m$ shows a gradual plateau during the charging phase, indicating that $V_m$ can only reach $V_{th}$ after $V_g$ has decreased enough.

so large that the reduced input current cannot charge $C_1$ sufficiently to raise $V_m$ to $V_{th}$, therefore the neuron stops firing. Thus, is it evident that the spiking frequency of the proposed ALIF neuron can be modulated by $V_g$. The effect of the width-to-length (*W/L*) ratio of $M_3$ on the spiking frequency was also simulated and the results are plotted in Supplementary Fig. 13, showing that, for a given $V_g$, a larger *W/L* ratio results in a lower spiking frequency due to an increased leakage current.

Next, we simulated the dynamical adaptation of the circuit by applying a constant step input current while allowing $V_g$ to dynamically evolve. The resulting waveforms of $V_g$, $V_m$, and $V_{spike}$ are illustrated in Fig. 3d. The evolution of the *ISI* is illustrated in the curve corresponding to the controlled condition in Supplementary Fig. 15. As $V_g$ increases with each spike, the spiking frequency remains constantly high initially, before decreasing at an increasing rate, which is a similar trend as that in Fig. 3c. Eventually, $V_g$ is high enough such that $V_m$ cannot reach $V_{th}$ unless $V_g$ has decreased sufficiently. This is evident from the gradual plateau feature during the late charging phase in the $V_m$ waveform. As a result, $V_g$ simply oscillates around a fixed value with a prolonged period and the spiking frequency saturates at its lowest level. When the input signal is removed, $V_g$ decays at a rate much lower than that of $V_m$, illustrating the difference in their time constants. The adaptive property of the circuit can be tuned by adopting different values for *W/L* ($M_2$, $M_3$), $C_2$, and $R_3$, as elucidated in Supplementary Fig. 14-15. It can be seen that the onset of adaptation is later if either $C_2$ is large or the *W/L* of $M_2$ is small, as a larger capacitor and a smaller current result in a slower charging process. On the other hand, the saturation frequency can be tuned by $M_2$, $M_3$, and $R_3$. A larger *W/L* of $M_2$ induces a larger step increase in $V_g$, which dwells on its raised value longer if $R_3$ is larger, hence contributing to a larger *ISI*. Besides, $M_3$ with

a larger *W/L* requires a lower $V_g$ to achieve the same leakage current, and a lower $V_g$ decays at a slower rate, which increases the *ISI*. It is worth noting that the initial high frequency cannot be adjusted as it is solely determined by the LIF part. Thus, we presented the various tuning knobs to obtain different adaptive properties, which can be useful in optimizing the performance of the ALIF neurons. The benchmark of the ALIF neuron in this work against previous implementations is shown in Supplementary Table 3, highlighting the simplicity of our circuit.

### The asynchronous spike encoder based on VO₂ memristor

Another key role in neuromorphic physiological signal processing systems is the spike encoder. An ideal encoder should provide a compressed representation of the data while preserving as much information as possible[52]. In ordinary neurons, analog signals are encoded as spike frequencies which do not contain accurate timing information, making it difficult to reconstruct the original analog signals. The asynchronous spike encoder based on VO₂ memristor converts the input analog signal into two spike trains, a positive and a negative (UP/DOWN channel). The positive spike represents the moment when the input signal increases beyond a threshold, while the negative spike represents the moment when the input signal decreases beyond a threshold. The spike trains can accurately reconstruct the original input analog signal due to the inclusion of precise time information. The schematic of memristor-based asynchronous spike encoder is depicted in Fig. 4a, including input amplifier with a capacitive-divider gain stage, intermediate amplifier, VO₂ memristor and feedback reset branch. According to the law of conservation of charge at the negative input terminal node of the input stage op-amp,

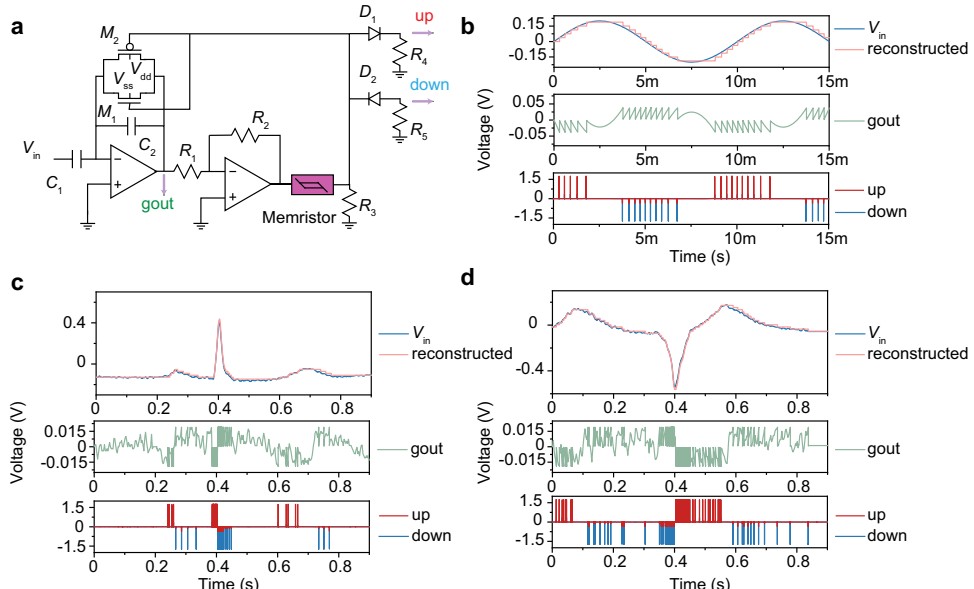

**Fig. 4 | Proposed memristor-based asynchronous spike encoder. a** Schematic of memristor-based asynchronous spike encoder which includes an input amplifier with capacitive-divider gain stage, an intermediate amplifier, a VO$_2$ memristor, and a feedback reset branch. The voltage at gout changes only when the input voltage changes which is then amplified by the op-amp in the middle stage and applied to the VO$_2$ memristor. Once the voltage of VO$_2$ memristor exceeds its positive and negative threshold voltage, VO$_2$ memristor will switch to low resistance and generate a large current. This current is converted to a high voltage by $R_3$, which will turn on the PMOS/NMOS transistor on the feedback path, thereby resetting gout. At this time, the voltage on the VO$_2$ memristor is lower than its holding voltage, and VO$_2$ memristor switches to high resistance state, so that the voltage of $R_3$ decreases and the PMOS/NMOS will be turned off. Finally, the positive and negative spikes are divided into two channels through two antiparallel diodes. A good reconstruction

of the original input signal can be achieved using the encoded spikes since it marks the time at which the input signal increases or decreases by a fixed value $\delta$. With this type of encoding, the spikes are sparse and the encoding is on-demand, which is friendly to neuromorphic systems. **b** The simulation results of memristor-based asynchronous spike encoder in LTSPICE when a sine signal is applied. The first row represents the original input signal (blue curve) and the reconstructed signal (pink curve). The second and third rows represent the voltage of the node gout and the output spike on UP/DOWN channel, respectively. Where the signal changes more violently, the spikes are more intensive. When the input signal increases by a fixed value $\delta$, a spike appears on the UP channel. Conversely, at the moment a spike appears on the DOWN channel, the input signal decreases by $\delta$. **c, d** The simulation results when two typical ECG waveforms are input to an asynchronous spike encoder.

the voltage at gout changes only when the input voltage changes. Then the voltage of gout is amplified by the intermediate stage op-amp and applied to the VO$_2$ memristor. When the voltage exceeds the positive/negative threshold $V_{th}$ of the memristor, the memristor will become LRS, thus issuing a positive/negative high voltage on $R_3$. Then, the high voltage will turn on the NMOS/PMOS through the feedback path, and reset the voltage of gout. At this moment, the voltage on the VO$_2$ memristor will be lower than positive/negative $V_{hold}$, thus the memristor will automatically return to HRS, the voltage on $R_3$ falls, thereby turning off the PMOS/NMOS on the feedback path. On the other hand, the positive and negative spikes are separated by two diodes to UP/DOWN channels as outputs. The threshold $\delta$ represents the incremental or decremental change of the input signal that causes a single spike, which can be described by Eq. 1 when $C_1 = C_2$:

$$\delta = \frac{V_{th}}{\alpha \frac{R_{off}}{R_{off} + R_3}} \tag{1}$$

where $V_{th}$ is the threshold voltage of the VO$_2$ memristor, while $R_{off}$ is the resistance of the HRS. $\alpha$ is the absolute amplification factor of the intermediate stage op-amp, which can be described by Eq. 2:

$$\alpha = \frac{R_2}{R_1} \tag{2}$$

It can be seen from the above formula that $\delta$ can be adjusted by the amplification factor $\alpha$ of the intermediate stage op-amp.

We first use a sine wave to verify the functionality of the memristor-based asynchronous spike encoder. Figure 4b exhibits the simulation results in LTSPICE, where the blue curve represents the original input, and the pink curve represents the reconstructed result, in the first row. It can be seen that the signal is well reconstructed. The middle row shows the node gout of the asynchronous spike encoder (green curve), which is next amplified by intermediate stage op-amp and applied to the VO$_2$ memristor. When the amplified voltage reaches the symmetrical positive/negative threshold voltage of the VO$_2$ memristor, the memristor switches to low resistance and emits a positive/negative spike, which is divided into two channels by two parallel reverse diodes as shown in the last row of Fig. 4b, where the red and blue curve represent the UP and DOWN channels, respectively. When a spike appears in the UP channel, it means that the original signal has increased by a $\delta$. On the contrary, when a spike appears in the DOWN channel, the original signal has decreased by a $\delta$. The frequency of spikes emission depends on the rate at which the original input signal changes. The faster the original input signal changes, the higher the intensity of the spikes. This type of encoding has the advantage of the sparsity of the spikes, and the on-demand nature of the encoding (when the input signal is not changing, no output spikes are produced)[11]. Supplementary Fig. 16 exhibits the influence of amplification factor $\alpha$ of the intermediate stage op-amp on the delta, the larger the $\alpha$, the smaller the $\delta$. Increasing the amplification factor improves the accuracy, but also increases the spike emission rate, thereby increasing energy consumption. We tested the encoder with two typical ECG signal waveforms, both of them can be well encoded and reconstructed as shown in Fig. 4c, d, demonstrating its potential as a next-generation neuromorphic spike encoder for physiological

signals. Due to the full use of the positive and negative thresholds and volatile characteristics of the $VO_2$ memristor, our circuit has been greatly simplified without using complex control circuits and ADC/DAC, compared with previous work (Supplementary Table 4). The simulation parameters of the asynchronous spike encoder are provided in Supplementary Table 5.

A key aspect that needs to be considered when using a $VO_2$ memristor in the asynchronous spike encoder is its reliability in encoding physiological signals. This can be assessed based on the lifespan of the encoder and the signal encoding quality. As aforementioned, the $VO_2$ memristor has a high endurance (Supplementary Fig. 6), which will ensure the durability of the encoder. On the other hand, the quality of signal encoding is affected by $V_{th}$ fluctuations. We introduced varying degrees of $V_{th}$ fluctuations in the SPICE model and performed multiple noisy encoding processes (Supplementary Note 3). The encoding quality was quantified by the mean squared error (MSE) between the original and the reconstructed signals. The results are plotted in Supplementary Fig. 17, along with examples of signal reconstruction under zero, moderate and high degrees of $V_{th}$ fluctuations. As the MSE and $C_v$ correlate positively, our $VO_2$ memristor, which has a remarkably low $C_v$, will yield accurately encoded spike outputs (Supplementary Note 3). Moreover, the tight MSE distribution at such low $C_v$ will also enable superior repeatability in spike encoding. Therefore, these results on the endurance and signal encoding quality attest to the reliability of our $VO_2$ memristor-based spike encoding architecture.

## $VO_2$ memristor-based LSNN and arrhythmia classification

Using the key modules presented above, we designed a robust and efficient physiological signal processing system with great temporal processing capacity. As shown in Fig. 5a, the system consists of two stages, namely the $VO_2$ memristor-based encoder followed by the decision-making $VO_2$-based LSNN[32]. The encoder converts analog physiological signals, for instance an ECG signal of a heartbeat, into UP and DOWN spike trains on a per input channel basis. These spike trains, which faithfully represent the original signal, are then relayed to the LSNN. Spatially, the LSNN is a 3-layer network comprising an input spiking layer, a hidden recurrent spiking layer with a low-pass filter, and an output classification layer. Each synaptic weight in all connections is assigned a random synaptic delay at network initialization. The core of the LSNN is the hidden recurrent layer, which consists of LIF neurons and ALIF neurons. It is the adaptive property of the ALIF neurons that endows LSNN with its temporal computing capability[51]. Here, we utilized the $VO_2$ memristor-based LIF neurons and ALIF neurons from the previous sections. During network simulations, the dynamics of these neurons were modeled according to their corresponding circuit designs and have the following discretized membrane dynamics equations (Eqs. 3–4):

$$V_{m\_LIF}(t + \Delta t) = \alpha V_{m\_LIF}(t) + (1 - \alpha)R_{eff}x \quad (3)$$

$$V_{m\_ALIF}(t + \Delta t) = \alpha V_{m\_ALIF}(t) + (1 - \alpha)R_{eff}(x - I_{leak}) \quad (4)$$

where $\alpha = \exp(-\Delta t/R_{eff}C_1)$. $R_{eff}$, $x$, and $I_{leak}$ are the effective resistance of the $VO_2$ memristor in HRS in series with the readout resistor, the input current scaled by a factor, and the leakage current via transistor $M_3$ due to adaptation, respectively. When $V_m$ exceeds $V_{th\_eff}$, it is reset to $V_{hold\_eff}$, where $V_{th\_eff}$ and $V_{hold\_eff}$ are the effective threshold and holding voltages of $VO_2$ memristor considering the readout resistor, respectively. $I_{leak}$ depends on $V_g$, which evolves according to the discretized dynamics equation (Eq. 5):

$$V_g(t + \Delta t) = \beta V_g(t) + (1 - \beta)R_3 I_a z \quad (5)$$

where $\beta = \exp(-\Delta t/R_3 C_2)$, $z$ is 1 if a spike was fired and 0 otherwise, and $I_a$ is the adaptive charging current via $M_2$. The forward pass during both the training and testing phases, as well as the backward pass during only the training phase, is shown in the flow chart in Fig. 5b. In the forward pass, the input spike vector of the current timestep and the hidden spike vector of the previous timestep are linearly transformed by the forward weights and the recurrent weights, respectively. The resulting vectors are added together and then integrated in the hidden layer to produce a hidden spike vector, which is subsequently low-pass filtered before being linearly transformed by output weights into an output vector. The output node with the highest value in the last timestep corresponds to the classification result. In the backward pass, the total error consisting of the classification cross-entropy loss and a spike regularization term, which promotes sparse firing of the spiking neurons[32], is back-propagated to train the fully-connected weights. As each spiking neuron is effectively a non-differentiable step activation function, we used a surrogate derivative for gradient calculations[32,53] (see "Methods").

Next, we investigated the performance of the proposed physiological signal processing system on classifying heartbeats from the MIT-BIH arrhythmia database[54]. The ECG recordings were pre-processed and categorized according to the AAMI recommended classes[55] (see "Methods"). 2000 heartbeat samples were used as our dataset, which were randomly split into a training set of 1664 samples and a testing set of 336 samples.

For this 4-class heartbeat classification task, we used the proposed system with an LSNN of size $3 \times 100 \times 4$, in which out of the 100 hidden neurons, 60 were LIF neurons while the other 40 were ALIF neurons. The 3 input nodes correspond to UP, DOWN, and CUE channels, while the 4 output nodes correspond to the 4 classes of heartbeats. Other LSNN parameters are listed in Supplementary Table 6, wherein the parameters describing the $VO_2$ memristor were extracted from experimental data. The single-channel analog heartbeat is encoded into an UP spike train and a DOWN spike train using the $VO_2$ memristor-based encoder as shown in Fig. 5c. Also shown is the CUE channel, which fires constantly at the end of the heartbeat, prompting the LSNN to generate a valid classification output. We trained the LSNN for 150 training epochs. Figure 5d, e plot the spike raster of the LIF neurons and ALIF neurons, respectively, when the trained system was classifying a normal heartbeat (Fig. 5a inset, Fig. 5c). Figure 5f illustrates the $V_g$ evolution of five ALIF neurons. Each step increase in $V_g$ implies a spike being fired by that neuron in the previous timestep, hence the spiking activities and the patterns within can be easily observed. ALIF neurons that fire at a high rate during the heartbeat (timestep -500) could not fire easily during the output period (time-step >1000) by virtue of their adaptive property manifested here as a high $V_g$, while those initially inactive neurons fired at a high rate during the output period. This exemplifies the negative imprinting principle[51], which equips the ALIF neurons with remarkable temporal processing capabilities. As can be seen from the output probabilities in Fig. 5g, the system correctly classified this heartbeat. The evolution of the test accuracy of the system is shown in Fig. 5h, indicating a maximum accuracy of 95.83%. The confusion matrix in Fig. 5i further illustrates the classification results in detail.

To further investigate the computational advantage of the $VO_2$ memristor-based ALIF neurons, we trained two other same-sized LSNNs but with different configurations, one with only hidden LIF neurons (LIF-only LSNN) while the other with only hidden ALIF (ALIF-only LSNN) neurons, on the same task. The spike raster plots, $V_g$ evolution, and output probabilities for an instance of a normal heartbeat are shown in Supplementary Figs. 18–19. The accuracy evolutions, loss evolutions, and confusion matrices are compared to the LSNN with both types of spiking neurons (mixed LSNN) as shown in Supplementary Fig. 20. We also trained the three configurations of LSNNs in

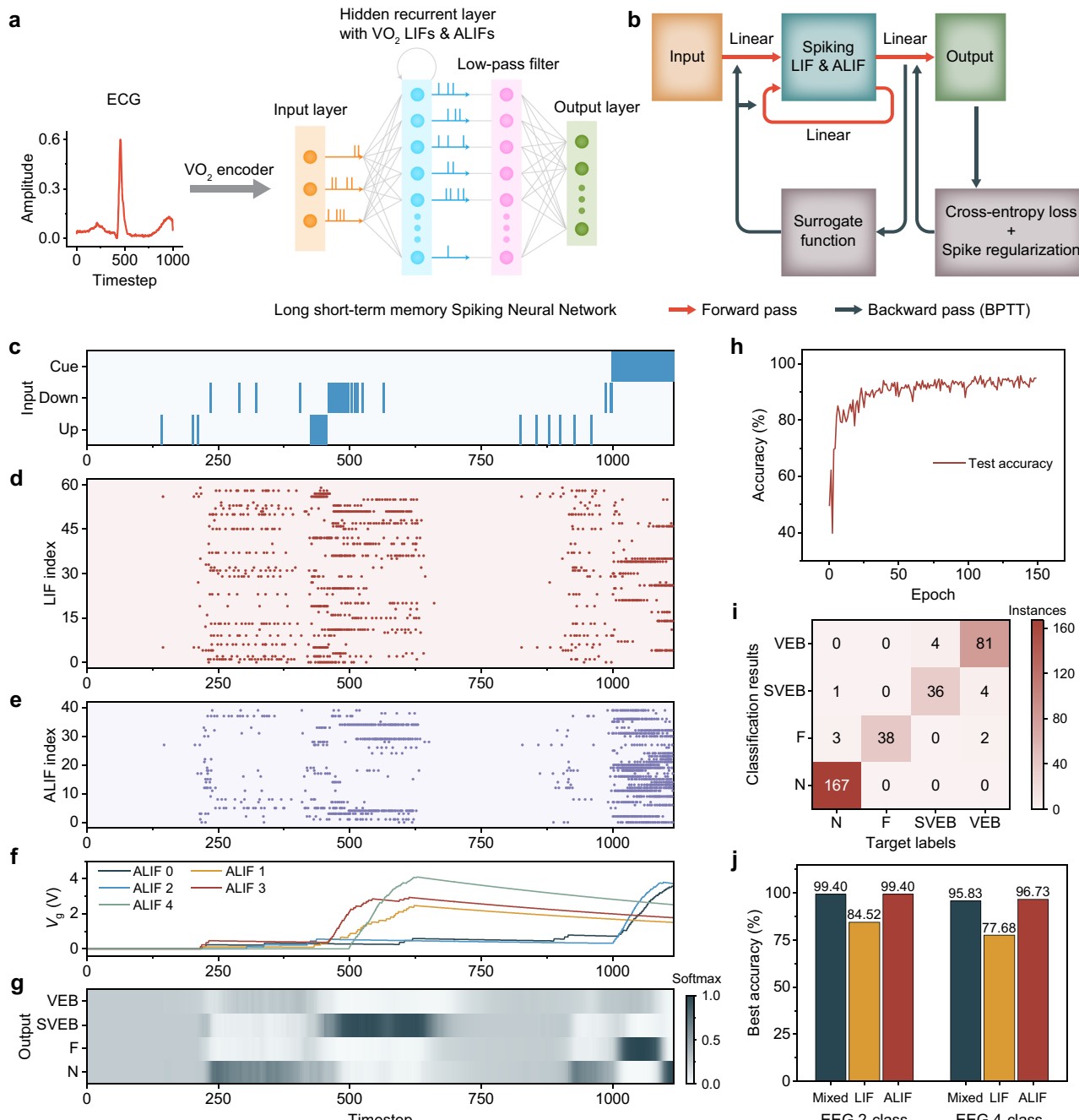

**Fig. 5 | Illustration of the VO$_2$ memristor-based LSNN in the physiological processing system for ECG heartbeat classification. a** Schematic of the VO$_2$ memristor-based physiological processing system detailing the LSNN decision-making stage, which is comprised of an input layer, a hidden recurrent layer with LIF neurons and ALIF neurons, a low-pass filter, and an output layer. The analog ECG heartbeat is first encoded by the VO$_2$ encoder into UP and DOWN spike trains, which are then relayed to the LSNN for classification. **b** Flow chart of the LSNN operation. In the forward pass, the weighted input spikes are added to the weighted hidden spikes from the previous timestep and fed into the hidden layer. The resulting hidden spikes are low-pass filtered and then weighted to generate output vectors. The output node with the highest value in the last timestep corresponds to the classification result. During back-propagation through time (BPTT), the cross-entropy loss and the spike regularization term are added together and propagated

backward for weight updates. A differentiable surrogate function is used to calculate the gradients. **c** Spike raster of the encoded normal heartbeat in (**a**) with a CUE signal prompting a valid output period. **d** Spike raster of the LIF neurons. **e** Spike raster of the ALIF neurons. **f** The $V_g$ evolution of five ALIF neurons showing that inactive (active) neurons before the output period fire with ease (difficulty) during the output period. **g** The evolution of output probabilities showing that the system correctly classified the normal heartbeat. **h** Evolution of test accuracy during 150 training epochs, reaching a maximum of 95.83%. **i** Confusion matrix of the classification results showing that the heartbeats were well classified. **j** Comparison of the maximum accuracies of the mixed LSNN, the LIF-only LSNN, and the ALIF-only LSNN in both the 2-class (N, not N) and the 4-class (N, VEB, SVEB, F) heartbeat classification tasks, illustrating the importance and the computing capacity of the ALIF neurons.

classifying heartbeats as Normal (N) or not Normal (not N). The LSNNs were of size $3 \times 20 \times 2$, wherein 8 out of the 20 hidden neurons in the mixed LSNN were ALIF neurons. The spike raster plots, $V_g$ evolution, and output probabilities for an instance of a normal heartbeat are shown in Supplementary Figs. 21–23, while the accuracy evolutions, loss evolutions, and confusion matrices are shown in Supplementary Fig. 24. The best test accuracy statistics for 18 training trials are shown in Supplementary Table 7. By comparing the maximum test accuracies attained by the three LSNN configurations as shown in Fig. 5j, two important observations can be made. Firstly, the LIF-only LSNN performed worse than the mixed LSNN by approximately 15% and 18% in terms of accuracy in the 2-class and the 4-class task, respectively, thereby highlighting the importance of ALIF neurons in processing the temporally-structured information within physiological signals. Secondly, the ALIF-only LSNN performed only marginally better than the mixed LSNN in general, thereby signifying the necessity of setting only a fraction of the hidden nodes as ALIF neurons to achieve superior performance. These findings elucidate the immense temporal computing capability of these neurons. Moreover, the design choice of using the mixed LSNN configuration is further justified by the potential reduction of area costs over the ALIF-only LSNN, especially when scaling up the system and considering its major use case in compact wearable medical devices.

### Epileptic seizure detection

To further verify the ability of the proposed $VO_2$ memristor-based physiological signal processing system in dealing with other complex signals, we demonstrated epileptic seizure detection on EEG signals from the CHB-MIT scalp EEG database[56]. The EEG recordings were preprocessed as detailed in the "Methods" section. 2530 EEG clips were used for training, which is comprised of 1265 Normal (N, negative class) and 1265 Epileptic (E, positive class) independent non-contiguous EEG clips. The testing set is comprised of 2878 contiguous EEG clips amounting to a one-hour period and is a highly imbalanced dataset with only 31 contiguous epileptic clips. The choice of a contiguous imbalanced testing set is to closely simulate a real-world scenario during epileptic seizures, wherein the seizure episodes are often sparse with each lasting for a short period of time. This is to ensure that our trained system can be deployed in real-time epileptic seizure detection in the future.

A schematic illustrating the training, testing, and post-processing steps of the epileptic detection system is shown in Fig. 6a. The 18-channel EEG clip (inset is an example of an epileptic clip) is first encoded by the $VO_2$ memristor-based encoder into 18 pairs of UP and DOWN spike trains, before being input into the LSNN. The training and testing phases of the LSNN are similar to that of the ECG task, wherein the encoded signal is classified by the LSNN in the forward pass during both phases, and the error, which includes cross-entropy loss and spike regularization, is back-propagated through time to update the weights during training only. Due to the highly imbalanced nature of the testing set, the performance of the LSNN during testing was evaluated using the G-mean metric[57], which is the geometric mean of the sensitivity and the specificity of the classification system (Eqs. 6–8):

$$\text{Sensitivity} = \frac{TP}{TP + FN} \qquad (6)$$

$$\text{Specificity} = \frac{TN}{TN + FP} \qquad (7)$$

$$G - \text{mean} = \sqrt{\text{Sensitivity} \cdot \text{Specificity}} \qquad (8)$$

where TP, FP, TN, and FN denote true positives, false positives, true negatives, and false negatives, respectively. To improve the system performance, especially in terms of specificity, a post-processing step was performed on the contiguous LSNN classification results to obtain the final classification results[58,59] (light purple box in Fig. 6a). It consists of a moving average operation, followed by a thresholding operation at each timestep to output a binary sequence, which is also contiguous in time. Note that the post-processing step is decoupled from the LSNN, and is not involved in loss calculation during training, or in model evaluation during testing.

The LSNN employed for this task was of size $37 \times 40 \times 2$, with 16 out of the 40 hidden spiking neurons being ALIF neurons. The 37 input nodes include 18 UP channels, 18 DOWN channels, and a CUE channel, while the 2 output nodes correspond to N and E. Other LSNN parameters are listed in Supplementary Table 6. We trained the LSNN for 150 epochs. The CUE signal and the encoded spike trains that represent the epileptic EEG clip in Fig. 6a are shown in Fig. 6b. The spike raster plots of the LIF and ALIF neurons when the trained LSNN was classifying this clip are shown in Fig. 6c, d, respectively. The $V_g$ evolution of five ALIF neurons is shown in Fig. 6e, while the output probabilities are shown in Fig. 6f. As can be seen, the system correctly classified this EEG clip. As shown in the confusion matrix in Fig. 6n, the accuracy, sensitivity, and specificity of the LSNN were 82.70%, 100%, and 82.51%, respectively. All of the positive epileptic EEG clips were accurately identified. We also trained a LIF-only LSNN and an ALIF-only LSNN on the same task (Supplementary Figs. 25–27), again corroborating the superior temporal processing capability of our ALIF neurons. The test G-mean statistics for 18 training trials are shown in Supplementary Table 8.

From these results, we can see that the specificity of the LSNN indicates a rather high number of false positives, possibly due to insufficient training data. The nature of the detected positives is revealed by visualizing the contiguous LSNN classification results (Fig. 6h) and comparing them against the target labels (Fig. 6g). While the accurately identified true positives spanned several contiguous EEG clips (Fig. 6k), the predicted false positives were randomly distributed throughout the one-hour period (Fig. 6l). This observation motivated the inclusion of the aforementioned post-processing step. As the moving average and the thresholding depends on the width of the averaging window and the threshold value, respectively, we optimized these parameters by enumerating their possible combinations and comparing the post-processing accuracies, sensitivities, and specificities (Supplementary Fig. 28). A window width of 9 and a threshold of 0.8 were selected for the best accuracy and sensitivity. As shown in Fig. 6i, the moving average smoothed out the random false positives while preserving the clustered true positives. Upon thresholding, the smoothened false positives were effectively removed (Fig. 6j), while the true positives were retained (Fig. 6m). As shown in the confusion matrix in Fig. 6o, the accuracy, sensitivity, and specificity after post-processing were 99.79%, 90.32%, and 99.89%, respectively. Owing to the efficient spike encoding scheme and the LSNN with high temporal processing capability, our system achieved state-of-the-art performance in various metrics while only needing 1–3 orders of magnitude fewer weights (Supplementary Table 9). The small network coupled with compact memristive circuit design for the encoder and the spiking neurons will benefit future hardware integration in biomedical devices and next-generation human-machine interfaces[60].

## Discussion

The proposed $VO_2$ memristor-based physiological signal processing system has a high area efficiency. To illustrate this, we compared each $VO_2$ circuit module with existing CMOS or memristor implementations (Supplementary Note 4). With proper device and circuit optimizations (Supplementary Table 10), the LIF and ALIF neuron can achieve a small area of ~41.3 $\mu m^2$ and ~53.4 $\mu m^2$, respectively. Besides achieving the smallest area overhead, it is worth noting that the optimized $VO_2$ memristor-based ALIF neuron is also superior in terms of the

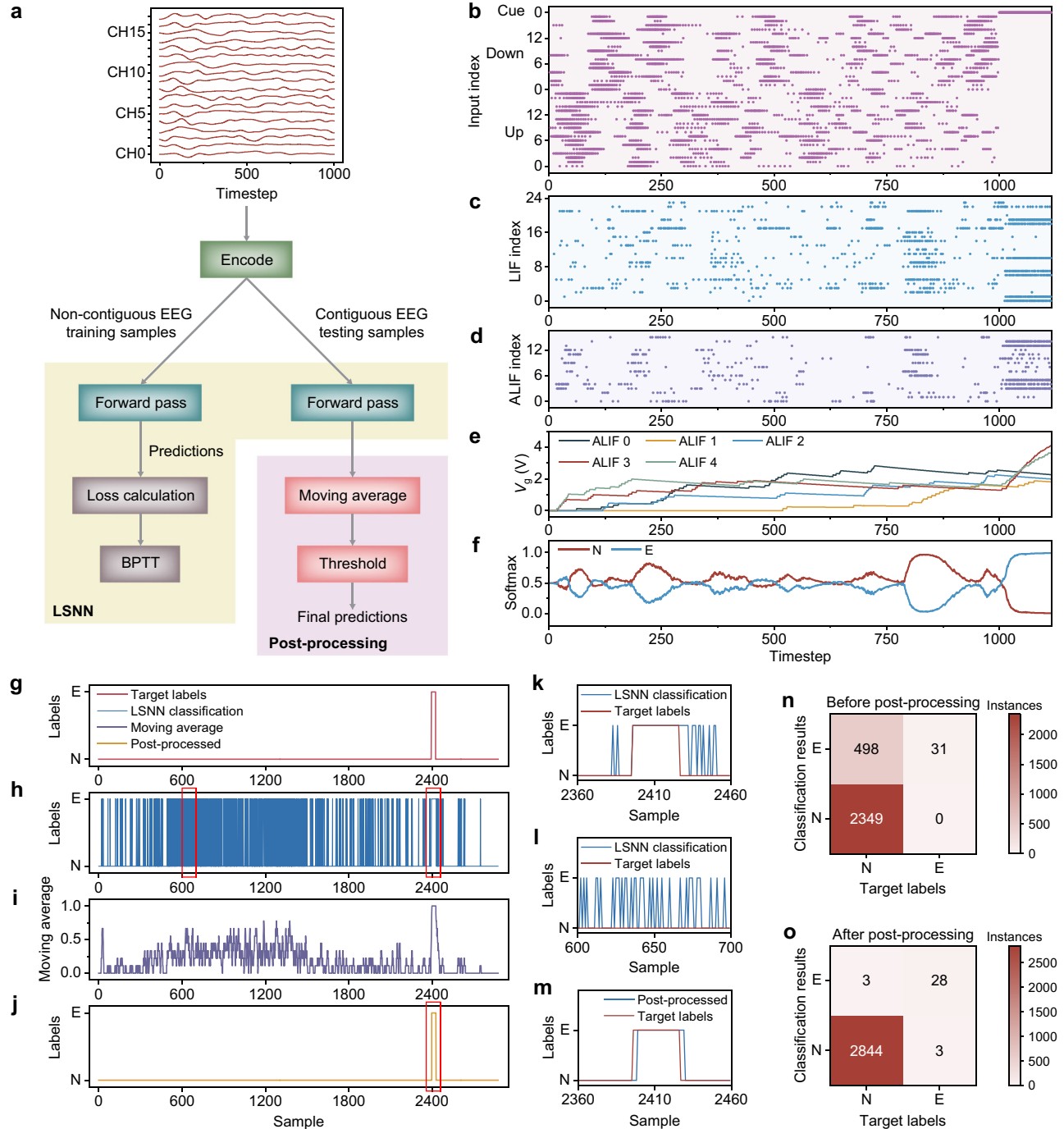

**Fig. 6 | Illustration of the physiological processing system for EEG seizure detection. a** Flow chart of the $VO_2$ memristor-based physiological signal processing system for epileptic seizure detection. Each EEG clip is encoded by the $VO_2$ encoder and then classified by the LSNN. During training only, the cross-entropy loss and the spike regularization term are calculated and propagated backward to update the weights. A final post-processing step consisting of moving average and thresholding improves the performance of the system. **b** Spike raster of the encoded epileptic EEG clip in (**a**) with a CUE signal prompting a valid output period. **c** Spike raster of the LIF neurons. **d** Spike raster of the ALIF neurons. **e** The $V_g$ evolution of five ALIF neurons. **f** The evolution of output probabilities showing that the system correctly classified the epileptic EEG clip. **g** Visualization of the contiguous target labels. **h** Visualization of the contiguous classification results of the

LSNN. **i** The results obtained after applying moving average with a window of 9, showing reduced values for regions with false positives, while maintaining high values for regions with true positives. **j** The final classification results after thresholding by a value of 0.8. **k** A zoomed-in plot of the area marked by the right red rectangle in (**h**), showing the contiguous nature of true positives. **l** A zoomed-in plot of the area marked by the left red rectangle in (**h**), showing the distributed nature of false positives. **m** A zoomed-in plot of the area marked by the red rectangle in (**j**), showing the correctly predicted epileptic seizure clips. **n** Confusion matrix of the raw LSNN classification results before post-processing. **o** Confusion matrix of the post-processed classification results, illustrating the benefits of the post-processing step in increasing specificity and accuracy.

combined aspects of area, speed and energy consumption (Supplementary Fig. 29, Supplementary Table 11). Furthermore, the proposed $VO_2$ memristor-based encoder can achieve an area of ~2231 $\mu m^2$, which is almost an order of magnitude smaller than other similar encoders (Supplementary Tables 12–13). Thus, $VO_2$ memristor-based encoder and neurons can provide substantial benefits over other CMOS or memristor implementations in realizing physiological signal processing systems. Further shrinking of $VO_2$ memristors is desirable in realizing hardware-based neural networks with an even higher integration density, especially in neuron circuits when capacitors, which are the dominant area-consuming components, are reduced or even replaced by the intrinsic parasitic capacitance for faster computations. Planar devices with gap sizes of 100 nm or less have been reported previously[40,61], and aggressive scaling down to the limits of lithography is possible given that the metal-insulator transition and, subsequently, the threshold switching behavior still exists at the nanoscale[62,63]. Apart from illustrating the benefits of our proposed physiological signal processing system, another takeaway from this discussion is the need for meticulous co-optimizations between various circuit components. The demonstrated co-optimizations, although simple, represent the first of many steps that need to be emphasized. Lastly, we further envision the merging of the $VO_2$ memristor-based encoders and neurons with non-volatile crossbar arrays of emerging memories[27] via proper interfacing (Supplementary Fig. 30) to ultimately realize an extremely compact physiological signal processing architecture.

In summary, a highly efficient neuromorphic physiological signal processing hardware system for the next-generation human-machine interface based on $VO_2$ memristors was proposed for the first time. This system contains a memristor-based asynchronous spike encoder and a decision network that features a long short-term memory spiking neural network which analyzes the physiological signals encoded in spikes. The spikes from memristor-based encoder mark the time at which the input signal has changed beyond a fixed threshold, which can preserve the original input information content to the greatest extent, so the encoded spikes can reconstruct the original signal accurately. The accuracy of signal encoding and reconstruction can be adjusted by the amplification factor of the intermediate stage op-amp. This spike encoding type has the advantage of sparse spikes and on-demand nature (no output spikes if input signal does not change). The asynchronous spike encoder was achieved efficiently without ADC/DAC and special control circuits due to the positive/negative symmetric threshold and volatile characteristics of the $VO_2$ memristor. In the decision-making LSNN, the ALIF neuron plays a key role which was achieved efficiently with $VO_2$ memristor. The release of each spike will change the current in the discharge path by feedback, achieving self-adaptation. The incorporation of ALIF neurons significantly improved the accuracy of the LSNN. The neuromorphic physiological signal processing system based on memristor achieved high accuracies of 95.83% and 99.79% with a very small LSNN on arrhythmia classification and epileptic seizure detection tasks, respectively. Our work demonstrated the potential and high efficiency of memristor-based neuromorphic systems for physiological signal processing, facilitating the construction of next-generation human-machine interfaces.

## Methods

### Fabrication of $VO_2$ memristor devices
The 20 nm $VO_2$ films were epitaxially grown on c-$Al_2O_3$ substrates by pulsed-laser deposition (PLD) technique using a 308-nm XeCl excimer laser operated at an energy density of about 1 J cm$^{-2}$ and a repetition rate of 3 Hz. The $VO_2$ films were deposited at 530 °C in a flowing oxygen atmosphere at the oxygen pressure of 2.0 Pa. Then, the films were cooled down to room temperature at the speed of 20 °C min$^{-1}$. The deposition rate of $VO_2$ thin films was calibrated by X-ray Reflection (XRR).

The $VO_2$ memristor was designed as a planar structure with a channel length of 400 nm and a width of 2 $\mu$m. The electrodes, which are composed of Au (40 nm) and Ti (5 nm) with a distance of 400 nm, were patterned with electron beam lithography (EBL) along with electron beam evaporation and lift-off.

### Electrical measurements
The $VO_2$ memristor was placed in a Signatone probe station to facilitate connections to the external circuit, source measurement unit and oscilloscope. As for measurements under various ambient pressures and in an $N_2$ environment in Supplementary Fig. 5, the $VO_2$ memristor was placed in a LakeShore cryogenic probe station. Electrical measurements were performed using an Agilent B1500A semiconductor parameter analyzer and the RIGOL MSO8104 digital storage oscilloscope. We used an Agilent B1500A semiconductor parameter analyzer to perform electrical measurements of a single $VO_2$ device in Fig. 2b and Supplementary Fig. 5. In Supplementary Figs. 4, 6 and 7, Agilent B1500A was applied to create the pulse signal, and the oscilloscope was used to measure either the voltage across the device or the current on the device. The experimental setup depicted in Supplementary Fig. 8 was used to connect the $VO_2$ device to the external LIF circuit for electrical measurements. In Fig. 2e–h and Supplementary Figs. 9–11, Agilent B1500A was applied to create the input signal, and the oscilloscope was used to measure the output of Agilent B1500A, the voltage on the capacitor and the output of the LIF neuron circuit.

### The physiological signal dataset
The MIT-BIH heart arrhythmia database[54] contains 30 min ECG recordings from 48 subjects. In order to improve the simulation accuracy, the original ECG waveforms were resampled at a frequency of 1800 Hz and split into single heartbeats of ~556 ms (1000 time-steps). Then, the heartbeats were normalized to 0–0.6 V. In total, 2000 different heartbeats were used as the dataset for this task, wherein the Normal (N), Ventricular ectopic beat (VEB), Supraventricular ectopic beat (SVEB), and Fusion beat (F) classes consisted of 1000, 500, 250, and 250 heartbeats, respectively.

The seizure data was obtained from CHB-MIT Scalp EEG Database[56]. The CHB-MIT database contains scalp EEG recordings from 22 patients at the Children's Hospital Boston. The original sampling rate of the database is 256 Hz. The EEG data in this work was from patient 1 where the data were resampled to 800 Hz. We selected 2530 data clips from the database for training. Each data clip contained 1000 time-step data with 18 channels (~1.25 s). The test sets were arranged in a one-hour-long segment to simulate the real-world situation. The EEG waveforms were all normalized to 0–0.6 V.

### The SPICE model of $VO_2$ memristor
The schematic diagram of the planar $VO_2$ memristor model is shown in Supplementary Fig. 12. First, the biasing polarity is determined. If $V_{top} \geq V_{bot}$, then the model compares $V_{top} - V_{bot}$ to the thresholds: in HRS, the model checks if $V_{top} - V_{bot} \geq V_{th}$ (equivalently $V_{top} \geq V_{bot} + V_{th}$), and in LRS, the model checks if $V_{top} - V_{bot} \leq V_{hold}$ (equivalently $V_{top} \leq V_{bot} + V_{hold}$). If $V_{top} < V_{bot}$, the model compares $V_{bot} - V_{top}$ to the thresholds, that is, checking $V_{bot} \geq V_{top} + V_{th}$ and $V_{bot} \leq V_{top} + V_{hold}$ in HRS and LRS, respectively. The right-hand sides of these four inequalities are constructed using the four voltage sources on the left to give $V_{top}^{+}$ and $V_{bot}^{+}$, taking into account the state of the device given by $V_o$ (1 in HRS, 0 in LRS). These comparisons are done by a comparator, which is modeled here by the behavioral voltage source $V_o$ according to Eq. 9:

$$V_o = \frac{1}{2}\left[1 + \tanh(2\alpha\Delta V)\right] \qquad (9)$$

where $\Delta V$ is described by Eq. 10:

$$\Delta V = \begin{cases} V_{bot}^+ - V_{top}, & V_{top} \geq V_{bot} \\ V_{top}^+ - V_{bot}, & V_{top} < V_{bot} \end{cases} \quad (10)$$

This will result in a hysteretic $I$-$V$ behavior typical of such devices. To model the finite switching time, $R_O$ and $C_O$ are introduced to suppress instantaneous change in $V_o$. The resistance of the device, $R_{VO2}$, which is determined by $V_c$, is then given by Eq. 11:

$$\frac{1}{R_{VO_2}} = \frac{1 - V_c}{R_{off}} + \frac{V_c}{R_{on}} \quad (11)$$

## Simulation of the LSNN

The decision-making LSNN for physiological signal processing was implemented using the PyTorch-based SpikingJelly module[64]. BPTT was employed to train the network. The total loss $L$ is given by Eq. 12:

$$L = \frac{1}{B}\sum_{i=1}^{B}\sum_{j=1}^{C} -t_{ij}\log\sigma\left(y_{ij}\right) + \frac{\lambda_f}{N}\sum_{n=1}^{N}\left(\bar{f}_n - f_0\right)^2 \quad (12)$$

The first term on the right-hand side is the categorical cross-entropy loss considering $C$ number of categories and a training batch size of $B$. $y_{ij}$ and $t_{ij}$ are the raw output (logits) and the target output, respectively, of the $j$-th output neuron for the $i$-th input sample. $\sigma(\cdot)$ is the softmax function. The second term describes the spike regularization for sparse firing[32], which is the mean squared difference between the average firing rate of each hidden neuron and the target frequency $f_0$. $N$ is the total number of hidden neurons and $\lambda_f$ is the regularization coefficient. The average firing rate of the $n$-th neuron is given by Eq. 13:

$$\bar{f}_n = \frac{1}{\Delta t} \cdot \frac{1}{B}\sum_{i=1}^{B}\left(\frac{1}{T}\sum_{j=1}^{T} z_{ijn}\right) \quad (13)$$

where $T$ is the total number of timesteps, $\Delta t$ is the length of each timestep, and $z_{ijn}$ is the presence of a spike at the $j$-th timestep during the $i$-th input sample. Spiking neurons can be regarded as having a non-differentiable step activation function, thus a surrogate derivative described by Eq. 14 was used for gradient calculations[32,53].

$$f(x) = \max[0, \gamma(1 - |x|)] \quad (14)$$

The values for all relevant parameters are listed in Supplementary Table 6.

## Data availability

All data supporting this study and its findings are available within the article, its Supplementary Information and associated files. The source data underlying Figs. 2b, c, e–l, 3c, d, 4b–d, 5a, c–j, 6a–o have been deposited in [https://zenodo.org/record/7888750] or are available from the corresponding author upon reasonable request.

## Code availability

The codes used for the simulations are described in [https://github.com/pekjuntiw/NCOMMS-23-03137] or are available from the corresponding author upon reasonable request.

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

## Acknowledgements

This work was supported by the National Natural Science Foundation of China (61925401, 92064004, 61927901, 92164302) and the 111 Project (B18001). Y.Y. acknowledges support from the Fok Ying-Tong Education Foundation and the Tencent Foundation through the XPLORER PRIZE.

## Author contributions

R.Y. and P.J.T. contributed equally to this work. R.Y., C.L., and C.G. fabricated the VO₂ devices. R.Y., T.Z., and C.L. performed electrical measurements. R.Y., P. J.T., L.C., and Z.Y. performed the simulations. R.Y., P.J.T., and Y.Y. prepared the manuscript. Y.Y. and R.H. directed all the research. All authors analyzed the results and implications and commented on the manuscript at all stages.

## Competing interests

The authors declare no competing interests.
