## [Peer Review File · Nature Communications]

REVIEWER COMMENTS

Reviewer #1 (Remarks to the Author):

In this manuscript, Yuan et al. fabricate VO₂ memristors and use them to construct hardware for human-machine interface signal processing. I have seen other manuscripts using VO₂ memristors but, in my opinion, this manuscript goes beyond the state-of-the-art in that it achieves better performance and it proposes novel applications that could be realized realistically based on the performance of the devices.

In fact, the whole study is very complete because it deals with materials synthesis, device fabrication and characterization, SPICE modelling and neural network implementation. Moreover, the datasets used in the neural network part are original, I am not aware of other studies in the field of memristors employing this type of datasets, and the performance of the neural network is excellent.

Here there are a few minor questions that I have that I think should be clarified before the publication of this manuscript:

1 – The structures are planar. Why? Does this also work with vertical structure? Did the authors try?

2 – The probe station is working in air? If the structures are planar, how does the moisture on the samples affect the measurements?

3 – A discussion related to actual implementation of this type of planar memristors into real systems should be included, as neural networks for brain-inspired computer required the integration of multiples devices with a high integration density.

Other than this, in my opinion, this study deserves publication in Nature Communications because it provides significant advancement into the state of the art in terms of novelty (application) and performance. The paper is indeed defect free, and the explanations and figures are clear.

Reviewer #2 (Remarks to the Author):

1. The paper by Yuchao Yang et al reports on designing of a physiological signal processing hardware system for human-machine interface. The authors have fabricated a 20 nm thick VO₂ planar memristor device on a Al₂O₃ substrate with a channel length of 400 nm. By using a SPICE simulator biasing conditions are determined for memristor. Spike Jelly framework has been used for training the neural network. The authors demonstrated the development of physiological signal processing system with the use of VO₂ memristor and presented the manuscript in a good way. Overall, the novelty of work is very good. Usefulness of proposed architecture shows potential applications in brain inspired computing. The presented results are interesting and may be considered for publication in Nature Communications after a minor revision. The following comments should be addressed before it is considered for publication.

2. How a fabricated memristor device was implemented on hardware to external circuit?

3. How stable is device in terms of threshold voltage variations when it is connected to external circuits?

4. How reliable is the architecture in encoding the ECG/EEG signals by using a VO₂ as an active material?

5. The memristor behavior of VO₂ as well as the physiological signal processing using other CMOS devices and memristor are also known. Hence the author should clearly highlight the advantage of VO₂ based memristor for physiological signal processing in clear terms. The mentioned aspect of efficiency etc are not very clear and appeared to be very generic.

6. Although the application part is presented well, the scientific findings or mechanism part is not standing out. It will be good, if the author describes the mechanism of memristor with more details. For example, the mentioned phase transition induced filament formation do not have any direct evidence and needs to be strengthened.

MS No: NCOMMS-23-03137

Title: A neuromorphic physiological signal processing system based on VO₂ memristor for next-generation human-machine interface

Response to the editor and the reviewers

We would like to express our gratitude to the editor for the kind consideration of our manuscript and to the reviewers for their constructive feedbacks, which are extremely valuable in improving our manuscript. We have thoroughly considered all the reviewers' questions and have made corresponding revisions. Additional experiments and simulations (Supplementary Fig. 3-6, 17, 29, Supplementary Tables 10-13, Supplementary Notes 1-3) have been performed. We have also included necessary discussions (Supplementary Fig. 8, 30) in the manuscript. We hope that the editor and reviewers will find the revised manuscript suitable for publication in *Nature Communications*. The point-to-point responses and changes made are listed below.

Reviewer #1 (Remarks to the Author):

In this manuscript, Yuan et al. fabricate VO₂ memristors and use them to construct hardware for human-machine interface signal processing. I have seen other manuscripts using VO₂ memristors but, in my opinion, this manuscript goes beyond the state-of-the-art in that it achieves better performance and it proposes novel applications that could be realized realistically based on the performance of the devices.

In fact, the whole study is very complete because it deals with materials synthesis, device fabrication and characterization, SPICE modelling and neural network implementation. Moreover, the datasets used in the neural network part are original, I am not aware of other studies in the field of memristors employing this type of datasets, and the performance of the neural network is excellent.

Here there are a few minor questions that I have that I think should be clarified before the publication of this manuscript:

Our response: We would like to sincerely thank the reviewer for the positive remarks. We also truly appreciate the valuable comments and constructive suggestions by the reviewer and have performed additional experiments and simulations (Supplementary Fig. 4, 5, 17, Supplementary Tables 10-13, Supplementary Notes 1-3). Our point-to-point responses to each question and corresponding revisions made are as follows.

1 – The structures are planar. Why? Does this also work with vertical structure?

Did the authors try?

Our response: We would like to thank the reviewer for raising this question.

The main reason for using planar VO₂ devices is due to their outstanding cycle-to-cycle (C2C) uniformity. To understand how the planar structure allows for high uniformity, we must first consider the origin of device variations. The stochasticity of VO₂ stems from the existence of multiple domains in the film and the competition between these domains in nucleating and initiating filament formation^{1,2}, which means that stochasticity can be effectively suppressed by obtaining high crystallinity VO₂ films. High crystallinity epitaxial VO₂ can be grown on c-Al₂O₃ via pulsed laser deposition as evident from Supplementary Fig. 1 and is attributed to the matching lattice

planes across the film-substrate interface³. The fabrication of planar VO₂ devices only requires the deposition of electrodes after film growth, which does not compromise the excellent characteristics of the film-substrate interface and the film itself, therefore enabling low C2C variations as can be seen from Fig. 2b-c. To quantify such variations, we computed the coefficient of variation, which is defined by $C_v = \sigma/\mu$. The C_v of V_{th_pos} , V_{hold_pos} , V_{th_neg} , and V_{hold_neg} were 0.65%, 0.86%, 0.31% and 1.68%, respectively. In addition to the uniformity observed under steady state, the VO₂ memristor also displayed very small variations in V_{th} and V_{hold} when it was connected to an external circuit and was operating in a dynamical state (Supplementary Fig. 4). The C_v of V_{th} and V_{hold} during transient oscillations were 0.73% and 0.48%, respectively. On the contrary, vertical VO₂ devices necessitate growing VO₂ on a bottom electrode and often result in polycrystalline or non-stoichiometric amorphous films^{4,5}, of which the latter requires additional post-annealing or electroforming procedures to form a polycrystalline channel. Therefore, the C2C variation in vertical devices is often larger than in epitaxial planar devices.

To exemplify the importance of C2C uniformity in our system, we further investigated the impact of V_{th} variations on the encoding quality of the asynchronous spike encoder. To emulate V_{th} fluctuations, we superimposed Gaussian noise with a zero mean and varying standard deviations ($\sigma = 0.00$ V, 0.01 V, 0.02 V, 0.03 V, 0.05 V, 0.08 V, 0.10 V, 0.13 V and 0.15 V) on the constant V_{th} in our VO₂ SPICE model. At a V_{th} of 3.4 V, this is equivalent to C_v values ranging from 0% to 4.41%. We randomly selected 8 ECG signals and simulated the noisy encoding process ten times for every combination of signal and σ , totaling 720 trials. In each trial, different random noise was regenerated. The mean squared error (MSE) between the original and the reconstructed signal was calculated for each trial. The average MSE for each C_v and the

individual MSE of each trial are plotted in Supplementary Fig. 17a. Also indicated in the figure is the C_v of V_{th_pos} of our VO₂ device. In general, the MSE increases with C_v as a larger fluctuation degrades encoding quality. The distribution of the individual MSE also increases with C_v . Examples of the reconstructed signal, voltage across the VO₂ device and V_{th} for zero, moderate ($C_v = 2.35\%$) and high ($C_v = 4.41\%$) degrees of V_{th} fluctuation are plotted in Supplementary Fig. 17b-d, respectively. By inspecting the signal reconstruction in Supplementary Fig. 17c and comparing its MSE with the individual MSEs for $C_v < 0.88\%$ ($\sigma < 0.03$ V), we can infer that the proposed encoder based on our VO₂ memristor can perform accurate spike encoding. The tight MSE distribution at low C_v will also contribute to achieving superior repeatability in spike encoding.

Therefore, we opted for the more superior planar structure and did not try using vertical devices to ensure that physiological signal processing systems incorporating VO₂ devices can function reliably.

To address these questions, we have added the new experimental and simulation results in Supplementary Fig. 4 and 17. We have also included discussions in the main text of the revised manuscript along with Supplementary Notes 1 and 2:

- Page 8-9:

“Fig. 2c displays the cumulative plots of positive and negative threshold/holding voltages, including V_{th_pos} , V_{hold_pos} , V_{th_neg} , and V_{hold_neg} in 100 repeated cycles. The coefficient of variation (C_v) defined by the ratio of the standard deviation (σ) to the mean value (μ) of V_{th_pos} , V_{hold_pos} , V_{th_neg} , and V_{hold_neg} were 0.65%, 0.86%, 0.31% and 1.68%, respectively, showing very low cycle-to-cycle (C2C) variations. The superior uniformity can be attributed to the high crystallinity epitaxial VO₂ thanks to the

matching lattice planes across the film-substrate interface⁴⁸, as well as the preservation of such desirable qualities in a planar device structure (Supplementary Note 1). In addition to the uniformity observed under steady state, the VO₂ memristor also displayed very small variations in V_{th} and V_{hold} when it was connected to an external circuit and was operating in a dynamical state (Supplementary Fig. 4). The C_v of V_{th} and V_{hold} during ~1000 periods of transient oscillations were 0.73% and 0.48%, respectively.”

- Page 19-20:

“A key aspect that needs to be considered when using a VO₂ memristor in the asynchronous spike encoder is its reliability in encoding physiological signals. This can be assessed based on the lifespan of the encoder and the signal encoding quality. As aforementioned, the VO₂ memristor has a high endurance (Supplementary Fig. 6), which will ensure the durability of the encoder. On the other hand, the quality of signal encoding is affected by V_{th} fluctuations. We introduced varying degrees of V_{th} fluctuations in the SPICE model and performed multiple noisy encoding processes (Supplementary Note 2). The encoding quality was quantified by the mean squared error (MSE) between the original and the reconstructed signals. The results are plotted in Supplementary Fig. 17, along with examples of signal reconstruction under zero, moderate and high degrees of V_{th} fluctuations. As the MSE and C_v correlate positively, our VO₂ memristor, which has a remarkably low C_v , will yield accurately encoded spike outputs (Supplementary Note 2). Moreover, the tight MSE distribution at such low C_v will also enable superior repeatability in spike encoding. Therefore, these results on the endurance and signal encoding quality attest to the reliability of our VO₂ memristor-based spike encoding architecture.”

- Supplementary Note 1: The rationale behind choosing a planar structure for the VO₂ memristor

“The main reason for using planar VO₂ devices is due to their outstanding cycle-to-cycle (C2C) uniformity. To understand how the planar structure allows for high uniformity, we must first consider the origin of device variations. The stochasticity of VO₂ stems from the existence of multiple domains in the film and the competition between these domains in nucleating and initiating filament formation^{15,16}, which means that stochasticity can be effectively suppressed by obtaining high crystallinity VO₂ films. High crystallinity epitaxial VO₂ can be grown on c-Al₂O₃ via pulsed laser deposition as evident from Supplementary Fig. 1 and is attributed to the matching lattice planes across the film-substrate interface¹⁷. The fabrication of planar VO₂ devices only requires the deposition of electrodes after film growth, which does not compromise the excellent characteristics of the film-substrate interface and the film itself, therefore enabling low C2C variations as can be seen from Fig. 2b-c. To quantify such variations, we computed the coefficient of variation, which is defined by $C_v = \sigma/\mu$. The C_v of V_{th_pos} , V_{hold_pos} , V_{th_neg} , and V_{hold_neg} were 0.65%, 0.86%, 0.31% and 1.68%, respectively. In addition to the uniformity observed under steady state, the VO₂ memristor also displayed very small variations in V_{th} and V_{hold} when it was connected to an external circuit and was operating in a dynamical state (Supplementary Fig. 4). The C_v of V_{th} and V_{hold} during transient oscillations were 0.73% and 0.48%, respectively. On the contrary, vertical VO₂ devices necessitate growing VO₂ on a bottom electrode and often result in polycrystalline or non-stoichiometric amorphous films^{18,19}, of which the latter requires additional post-annealing or electroforming procedures to form a polycrystalline channel. Therefore, the C2C variation in vertical devices is often larger than in epitaxial planar devices.

The importance of C2C uniformity in our system is further exemplified by the results shown in Supplementary Fig. 17, wherein the impact of C2C variations in the form of V_{th} fluctuations on the encoding ability of the VO₂ memristor-based asynchronous spike encoder was studied (see also Supplementary Note 2 for details). Larger C2C variations in V_{th} was found to be detrimental to the encoding quality. Therefore, we opted for the more superior planar structure instead of the vertical structure to ensure that physiological signal processing systems incorporating VO₂ devices can function reliably.”

- Supplementary Note 2: The impact of V_{th} fluctuations on spike encoding quality

“To emulate V_{th} fluctuations, we superimposed Gaussian noise with a zero mean and varying standard deviations ($\sigma = 0.00 V, 0.01 V, 0.02 V, 0.03 V, 0.05 V, 0.08 V, 0.10 V, 0.13 V$ and $0.15 V$) on the constant V_{th} in our VO₂ SPICE model. At a V_{th} of $3.4 V$, this is equivalent to C_v values ranging from 0% to 4.41% . We randomly selected 8 ECG signals and simulated the noisy encoding process ten times for every combination of signal and σ , totaling 720 trials. In each trial, different random noise was regenerated. The mean squared error (MSE) between the original and the reconstructed signal was calculated for each trial.

The average MSE for each C_v and the individual MSE of each trial are plotted in Supplementary Fig. 17a. Also indicated in the figure is the C_v of V_{th_pos} of our VO₂ device. In general, the MSE increases with C_v as a larger fluctuation degrades encoding quality. The distribution of the individual MSE also increases with C_v . Examples of the reconstructed signal, voltage across the VO₂ device and V_{th} for zero, moderate ($C_v = 2.35\%$) and high ($C_v = 4.41\%$) degrees of V_{th} fluctuation are plotted in Supplementary Fig. 17b-d, respectively. By inspecting the signal reconstruction in Supplementary Fig.

17c and comparing its MSE with the individual MSEs for $C_v < 0.88\%$ ($\sigma < 0.03$ V), we can infer that the proposed encoder based on our VO₂ memristor can perform accurate spike encoding. The tight MSE distribution at low C_v will also contribute to achieving superior repeatability in spike encoding.”

Supplementary Figure 4. V_{th} and V_{hold} variations under dynamical conditions. (a)

The extraction of V_{th} and V_{hold} of a VO₂ memristor when it was connected to an external neuron circuit and undergoing oscillations. **(b)** The distribution of the extracted V_{th} and V_{hold} from ~ 1000 periods of oscillation illustrating very low variations.

Supplementary Figure 17. The influence of C_v of V_{th} on the quality of signal encoding. (a) A scatter plot of the individual MSE between the original and the reconstructed signals for each of the 720 encoding trials and the average MSE for each C_v . The dotted line indicates the C_v of V_{th_pos} of our VO₂ memristor. (b)-(d) The original ECG signal (upper panel, blue), reconstructed signal (upper panel, red), voltage across the VO₂ memristor (middle and lower panel, green), and V_{th} (middle and lower panel, orange) under (b) zero, (c) moderate and (d) high degrees of fluctuations.

2 – The probe station is working in air? If the structures are planar, how does the moisture on the samples affect the measurements?

Our response: We would like to thank the reviewer for raising this interesting question.

The probe station we used was indeed working in air. Therefore, the planar VO₂ devices were exposed to moisture, which could have affected our measurements. In light of this, we carried out additional experiments in which the threshold switching behavior of the VO₂ memristor was characterized in air under normal atmospheric pressure (Supplementary Fig. 5a), under different ambient pressures ranging from 3.5×10^{-3} mbar down to 5.0×10^{-4} mbar (Supplementary Fig. 5b-g) and in an N₂ environment (Supplementary Fig. 5h). By doing so, the ambient moisture content was progressively reduced. No appreciable difference in the *I-V* curves can be observed. The V_{th} and V_{hold} are also stable under different ambient pressures as shown in Supplementary Fig. 5i. Therefore, the planar VO₂ device exhibited stable threshold switching behavior and is insensitive to moisture content, which implies that our measurements were not affected by moisture. It is worth pointing out that these results also showcase the stability of planar VO₂ devices when subjected to other atmospheric content in general.

To address this question, we have included the results in Supplementary Fig. 5 and added a brief discussion in the main text of the revised manuscript:

- Page 9:

*“Moreover, when the planar VO₂ memristor was operated in air under normal atmospheric pressure, under different ambient pressures ranging from 3.5×10^{-3} mbar down to 5.0×10^{-4} mbar and in an N₂ environment, it also exhibited stable threshold switching behavior with no appreciable difference in its *I-V* characteristics (Supplementary Fig. 5). This implies that such devices are not affected by various atmospheric content such as moisture.”*

“As for measurements under various ambient pressures and in an N_2 environment in Supplementary Fig. 5, the VO_2 memristor was placed in a LakeShore cryogenic probe station.”

Supplementary Figure 5. Threshold switching behavior of the planar VO_2 device under different environments. The VO_2 device was operated (a) in air under normal atmospheric pressure, (b)-(g) under different ambient pressures ranging from 3.5×10^{-3} mbar down to 5.0×10^{-4} mbar and (h) in an N_2 environment. By doing so, the ambient moisture content was progressively reduced. No appreciable difference in the I - V curves can be observed. (i) The V_{th} and V_{hold} were also stable under different ambient pressures.

3 – A discussion related to actual implementation of this type of planar memristors into real systems should be included, as neural networks for brain-inspired computer required the integration of multiples devices with a high integration density.

Our response: We would like to thank the reviewer for the constructive suggestion.

We fully agree with the reviewer on the need for achieving a small footprint and, thus, neural networks with high integration density. Although in the current circuit design, the footprint is dominated by capacitors instead of the VO₂ device, the same cannot be said when capacitors are reduced or even replaced by the intrinsic parasitic capacitance for faster computations. We acknowledge that the planar VO₂ device used in this work is still relatively large when compared with state-of-the-art transistors or even non-volatile RRAMs. Nevertheless, the device dimensions can still be optimized. Planar devices with gap sizes of 100 nm or less have been reported previously^{6, 7}. Moreover, the metal-insulator transition in VO₂ still exists at the nanoscale^{8, 9}, which implies that planar VO₂ devices can be scaled down to the limits of lithography while still retaining their threshold switching properties. Thus, planar VO₂ memristors can have sizes that are comparable to, or even smaller than, those of transistors, and might not be the limiting factor in achieving high integration densities.

It is worth elaborating that we must consider the area utilization at the circuit level to realize densely integrated hardware-based neural networks. Hence, as detailed in Supplementary Note 3, we estimated the area of our VO₂ memristor-based encoder and ALIF neuron using an optimized memristor with parameters based on ref⁷ and circuit parameters listed in Supplementary Tables 10 and 12. The estimations are summarized

in Supplementary Tables 11 and 13. The optimized encoder and ALIF neuron occupy $\sim 2231 \text{ um}^2$ and $\sim 53.4 \text{ um}^2$, respectively, which are much smaller than various other implementations. We further envision an extremely compact physiological signal processing architecture utilizing our VO₂ memristor-based encoder and neurons as shown in Supplementary Fig. 30. Coupled with the use of non-volatile crossbar arrays of emerging memories¹⁰, neural networks with a high integration density can then be easily achieved.

To address this point, we included Supplementary Fig. 30 as well as Supplementary Tables 10-13. We also added relevant discussions in the main text and Supplementary Note 3 of the revised manuscript:

- Page 32-33:

“The proposed VO₂ memristor-based physiological signal processing system has a high area efficiency. To illustrate this, we compared each VO₂ circuit module with existing CMOS or memristor implementations (Supplementary Note 3). With proper device and circuit optimizations (Supplementary Table 10), the LIF and ALIF neuron can achieve a small area of $\sim 41.3 \text{ um}^2$ and $\sim 53.4 \text{ um}^2$, respectively. Besides achieving the smallest area overhead, it is worth noting that the optimized VO₂ memristor-based ALIF neuron is also superior in terms of the combined aspects of area, speed and energy consumption (Supplementary Fig. 29, Supplementary Table 11). Furthermore, the proposed VO₂ memristor-based encoder can achieve an area of $\sim 2231 \text{ um}^2$, which is almost an order of magnitude smaller than other similar encoders (Supplementary Table 12-13). Thus, VO₂ memristor-based encoder and neurons can provide substantial benefits over other CMOS or memristor implementations in realizing physiological signal processing systems. Further shrinking of VO₂ memristors is desirable in

realizing hardware-based neural networks with an even higher integration density, especially in neuron circuits when capacitors, which are the dominant area-consuming components, are reduced or even replaced by the intrinsic parasitic capacitance for faster computations. Planar devices with gap sizes of 100 nm or less have been reported previously^{40,61}, and aggressive scaling down to the limits of lithography is possible given that the metal-insulator transition and, subsequently, the threshold switching behavior still exists at the nanoscale^{62,63}. Apart from illustrating the benefits of our proposed physiological signal processing system, another takeaway from this discussion is the need for meticulous co-optimizations between various circuit components. The demonstrated co-optimizations, although simple, represent the first of many steps that need to be emphasized. Lastly, we further envision the merging of the VO₂ memristor-based encoders and neurons with non-volatile crossbar arrays of emerging memories²⁷ via proper interfacing (Supplementary Fig. 30) to ultimately realize an extremely compact physiological signal processing architecture.”

- Supplementary Note 3: A comparison between different implementations of neurons and asynchronous spike encoders

“The proposed VO₂ memristor-based physiological signal processing system has a high area efficiency. To illustrate this, we compared each VO₂ memristor-based module with various other CMOS and memristor implementations.

First, we estimated the area of the VO₂ memristor-based neurons by considering the size of each individual circuit component. After appropriate circuit-level optimizations (Supplementary Table 10), our LIF and ALIF neurons occupy areas of only ~41.3 μm^2 and ~53.4 μm^2 , respectively. These figures are three to four orders of magnitude better than those before optimizations. Our ALIF neuron also shows at least

~2.2× area gains when compared to other implementations (Supplementary Table 11). One of the optimizations is to substantially reduce the capacitors as they dominate the area of the neurons. This is possible owing to the ability of VO₂ in switching at ultrafast timescales²⁰. Besides, the large resistors in the ALIF neuron should be replaced by transistors operating as current source loads. For the sake of completeness, the speed of the ALIF neurons in terms of the spiking frequency as well as the energy consumption are also given in Supplementary Table 11 (see also Supplementary Fig. 29), wherein we considered an optimized VO₂ memristor with a lower V_{th} and a higher HRS based on ref³. We then performed a more comprehensive comparison between ALIF implementations based on a figure of merit (FOM) given by

$$FOM = \frac{f}{E \cdot A}$$

where f , E and A are the spiking frequency, energy per spike and area, respectively. As summarized in Supplementary Table 11, our VO₂ memristor-based ALIF neuron can also achieve the highest FOM after optimizations.

Next, we estimated the area of the VO₂ memristor-based asynchronous spike encoder. In this case, the area is dominated by the op-amps and the capacitors. Here, an optimized VO₂ memristor was again considered¹³. Taking into account the capability of the op-amp in driving the VO₂ load circuit, the area of the op-amp was assumed to be 725 μm^2 as reported in ref¹⁴. Furthermore, the capacitors were also reduced while maintaining the encoding functionality of the architecture. Along with the size of other individual circuit components (Supplementary Table 12), the area was estimated to be ~2231 μm^2 . As summarized in Supplementary Table 13, the optimized VO₂ memristor-based encoder is almost an order of magnitude more compact than other similar encoders.

Therefore, we conclude that with appropriate device and circuit optimizations, a physiological processing system that employs our VO₂ memristor-based encoder and neurons is superior to other CMOS or memristor implementations primarily in terms of area benefits.”

Supplementary Figure 30. An overview of a compact hardware-based spiking neural network. To achieve high-density integration, such a network may employ our VO₂ memristor-based asynchronous spike encoder to convert analog signals into spikes, as well as VO₂ memristor-based LIF and ALIF neurons to perform biologically plausible neural computations. Further area savings can be made by utilizing dense non-volatile crossbar arrays of emerging memories for synaptic computations.

Supplementary Table 10. Parameters of the optimized ALIF

V_{dd}	1.2 V
I_{in}	26 μ A
R_1	4 k Ω
R_2 (PMOS current source as load)	$L = 260$ nm, $W = 65$ nm, $V_{g,bias} = 0.6$ V
R_3 (NMOS current source as load)	$L = 1040$ nm, $W = 65$ nm, $V_{g,bias} = 0.4$ V
C_1	100 fF
C_2	30 fF
M_1	$L = 65$ nm, $W = 65$ nm
M_2	$L = 100$ nm, $W = 100$ nm
M_3	$L = 65$ nm, $W = 65$ nm

Supplementary Table 11. Detailed comparison between adaptive LIF circuits

	Area (μm^2)	Energy per spike (J)	Adapted frequency (Hz)	FOM (Hz/ $(\mu\text{m}^2 \cdot \text{J})$)	Notes
Indiveri et al. (2006) ¹	2573	900×10^{-12}	-	-	• CMOS-based
Indiveri (2007) ²	-	-	-	-	• CMOS-based
Ferreira et al. (2019) ³	120	3.6×10^{-15}	205×10^3	4.75×10^{17}	• CMOS-based
Ahmadi-Farsani et al. (2022) ⁴	863	140×10^{-12}	1×10^3	8.28×10^9	• CMOS-based
Wang et al. (2018) ⁵	~ 320	-	30×10^6	-	• Memristor-based • Area estimated by calculating capacitor area only and using a capacitance density of 2.5 fF/ μm^2 .
Shaban et al. (2021) ⁶	11435.7	158×10^{-12}	-	-	• Memristor-based • Energy based on reported values for TSMC 65 nm ASIC simulation and spike count for SMNIST.

This work	~224000	6.92×10^{-9}	100×10^3	6.45×10^7	 • Memristor-based • Unoptimized. • Area estimated using a capacitance density of $2.5 \text{ fF}/\mu\text{m}^2$.
	~53.4	338×10^{-15}	33.2×10^6	1.84×10^{18}	 • Memristor-based • Optimized. • VO_2 based on ref¹³. • Circuit based on PTM 65 nm MOSFETs. • Area estimated using a capacitance density of $2.5 \text{ fF}/\mu\text{m}^2$ and a sheet resistance of $690 \Omega/\square$.

Supplementary Table 12. Parameters of the optimized spike encoder

Electric source	$V_{\text{dd}} = 1.2 \text{ V}, V_{\text{ss}} = -1.2 \text{ V}$
M_1	$W = 23 \mu\text{m}, L = 2.5 \mu\text{m}$
M_2	$W = 13 \mu\text{m}, L = 2.5 \mu\text{m}$
D_1 (Diode-connected PMOS)	$W = 1.3 \mu\text{m}, L = 65 \text{ nm}$
D_2 (Diode-connected NMOS)	$W = 0.7 \mu\text{m}, L = 65 \text{ nm}$
C_1	800 fF
C_2	800 fF
R_1	500 Ω
R_2	16.5 k Ω
R_3	1 k Ω
R_4	100 k Ω
R_5	100 k Ω

Supplementary Table 13. Detailed comparison between spike encoders

	Area (μm^2)	Notes
Sharifshazileh et al. (2021) ⁷	~13000	 • CMOS-based • Area estimated from chip design layout provided
He et al. (2021) ⁸	~17618	 • CMOS-based

		 • Area estimated from chip photograph provided
This work	~2231	 • Memristor-based • Optimized. • VO₂ based on ref¹³. • Op-amp based on ref¹⁴. • Circuit based on PTM 65 nm MOSFETs. • Area estimated using a capacitance density of 2.5 fF/μm² and a sheet resistance of 690 Ω/□.

Other than this, in my opinion, this study deserves publication in Nature Communications because it provides significant advancement into the state of the art in terms of novelty (application) and performance. The paper is indeed defect free, and the explanations and figures are clear.

Our response: We would like to thank the reviewer once again for the positive assessment.

Reviewer #2 (Remarks to the Author):

1. The paper by Yuchao Yang et al reports on designing of a physiological signal processing hardware system for human-machine interface. The authors have fabricated a 20 nm thick VO₂ planar memristor device on a Al₂O₃ substrate with a channel length of 400 nm. By using a SPICE simulator biasing conditions are determined for memristor. Spike Jelly framework has been used for training the neural network. The authors demonstrated the development of physiological signal processing system with the use of VO₂ memristor and presented the manuscript in a good way. Overall, the novelty of work is very good. Usefulness of proposed architecture shows potential applications in brain inspired computing. The presented results are interesting and may be considered for publication in Nature Communications after a minor revision. The following comments should be addressed before it is considered for publication.

Our response: We would like to sincerely thank the reviewer for the positive evaluation of our work. We also deeply appreciate the valuable feedbacks and suggestions. We have performed additional experiments and simulations (Supplementary Fig. 3, 4, 6, 17, 29, Supplementary Tables 10-13, Supplementary Notes 2, 3). Our point-to-point responses to each question along with corresponding changes made are as follows.

2. How a fabricated memristor device was implemented on hardware to external circuit?

Our response: We would like to thank the reviewer for the question.

In this study, the LIF neuron circuit was demonstrated using the experimental setup depicted in Supplementary Fig. 8. The VO₂ memristor was placed in a probe station to

facilitate connections to the external circuit, which includes the source measurement unit and the oscilloscope. As for the ALIF neuron circuit and the asynchronous spike encoder, they were simulated using the developed SPICE model of the memristor instead.

We would like to further elaborate that in next-generation human-machine interfaces, the hardware integration of such systems is ultimately required. In our case, this involves meticulous co-optimizations between various circuit components, for instance by taking into account the threshold switching characteristics of the memristor and the choice of transistors in terms of process node and the supply voltage. Here, we demonstrate such optimizations in a simple manner by considering a VO₂ device with parameters based on ref⁷ and using a PTM 65 nm model. The circuit parameters are listed in Supplementary Tables 10 and 12. Although still far from realizing a real integrated system, this simple demonstration represents the first step that needs to be emphasized. From a hardware architecture perspective, proper interfacing between VO₂ memristor-based encoders and neurons with synaptic computation modules such as non-volatile crossbar arrays of emerging memories (Supplementary Fig. 30) are also required.

To address this question, we have included Supplementary Fig. 8 and 30, Supplementary Tables 10 and 12, as well as the following experimental methods and discussions in the revised manuscript:

- Page 32-33:

“The proposed VO₂ memristor-based physiological signal processing system has a high area efficiency. To illustrate this, we compared each VO₂ circuit module with existing CMOS or memristor implementations (Supplementary Note 3). With proper

device and circuit optimizations (Supplementary Table 10), the LIF and ALIF neuron can achieve a small area of $\sim 41.3 \text{ um}^2$ and $\sim 53.4 \text{ um}^2$, respectively. Besides achieving the smallest area overhead, it is worth noting that the optimized VO₂ memristor-based ALIF neuron is also superior in terms of the combined aspects of area, speed and energy consumption (Supplementary Fig. 29, Supplementary Table 11). Furthermore, the proposed VO₂ memristor-based encoder can achieve an area of $\sim 2231 \text{ um}^2$, which is almost an order of magnitude smaller than other similar encoders (Supplementary Table 12-13). Thus, VO₂ memristor-based encoder and neurons can provide substantial benefits over other CMOS or memristor implementations in realizing physiological signal processing systems. Further shrinking of VO₂ memristors is desirable in realizing hardware-based neural networks with an even higher integration density, especially in neuron circuits when capacitors, which are the dominant area-consuming components, are reduced or even replaced by the intrinsic parasitic capacitance for faster computations. Planar devices with gap sizes of 100 nm or less have been reported previously^{40,61}, and aggressive scaling down to the limits of lithography is possible given that the metal-insulator transition and, subsequently, the threshold switching behavior still exists at the nanoscale^{62,63}. Apart from illustrating the benefits of our proposed physiological signal processing system, another takeaway from this discussion is the need for meticulous co-optimizations between various circuit components. The demonstrated co-optimizations, although simple, represent the first of many steps that need to be emphasized. Lastly, we further envision the merging of the VO₂ memristor-based encoders and neurons with non-volatile crossbar arrays of emerging memories²⁷ via proper interfacing (Supplementary Fig. 30) to ultimately realize an extremely compact physiological signal processing architecture.”

- Page 35-36: Methods – Electrical measurements

“The VO₂ memristor was placed in a Signatone probe station to facilitate connections to the external circuit, source measurement unit and oscilloscope. As for measurements under various ambient pressures and in an N₂ environment in Supplementary Fig. 5, the VO₂ memristor was placed in a LakeShore cryogenic probe station.”

and

“The experimental setup depicted in Supplementary Fig. 8 was used to connect the VO₂ device to the external LIF circuit for electrical measurements.”

Supplementary Figure 8. Experimental setup to measure the behavior of the LIF neuron circuit. The VO₂ memristor was placed in a Signatone probe station to facilitate connections to the external circuit, which includes the Agilent B1500A source measurement unit and the RIGOL MSO8104 oscilloscope.

Supplementary Figure 30. An overview of a compact hardware-based spiking neural network. To achieve high-density integration, such a network may employ our VO₂ memristor-based asynchronous spike encoder to convert analog signals into spikes, as well as VO₂ memristor-based LIF and ALIF neurons to perform biologically plausible neural computations. Further area savings can be made by utilizing dense non-volatile crossbar arrays of emerging memories for synaptic computations.

Supplementary Table 10. Parameters of the optimized ALIF

V_{dd}	1.2 V
I_{in}	26 μ A
R_1	4 k Ω
R_2 (PMOS current source as load)	$L = 260$ nm, $W = 65$ nm, $V_{g,bias} = 0.6$ V
R_3 (NMOS current source as load)	$L = 1040$ nm, $W = 65$ nm, $V_{g,bias} = 0.4$ V
C_1	100 fF
C_2	30 fF
M_1	$L = 65$ nm, $W = 65$ nm
M_2	$L = 100$ nm, $W = 100$ nm
M_3	$L = 65$ nm, $W = 65$ nm

Supplementary Table 12. Parameters of the optimized spike encoder

Electric source	$V_{dd} = 1.2$ V, $V_{ss} = -1.2$ V
M_1	$W = 23$ μ m, $L = 2.5$ μ m
M_2	$W = 13$ μ m, $L = 2.5$ μ m
D_1 (Diode-connected PMOS)	$W = 1.3$ μ m, $L = 65$ nm
D_2 (Diode-connected NMOS)	$W = 0.7$ μ m, $L = 65$ nm
C_1	800 fF
C_2	800 fF
R_1	500 Ω
R_2	16.5 k Ω
R_3	1 k Ω
R_4	100 k Ω
R_5	100 k Ω

3. How stable is device in terms of threshold voltage variations when it is connected to external circuits?

Our response: We would like to thank the reviewer for raising this important question.

As such devices have to be connected to external circuits in order to provide the desired functionalities, device stability in a circuit during transient operations is extremely important. In light of this, we have performed additional experiments wherein the VO₂ memristor was connected to an external circuit and allowed to oscillate. The V_{th} and V_{hold} were extracted from ~1000 periods of oscillations. We found that the VO₂ memristor displayed very small variations despite operating in a dynamical state. (Supplementary Fig. 4). The coefficient of variation (C_v) defined by the ratio of the standard deviation (σ) to the mean value (μ) of V_{th} and V_{hold} during transient oscillations were 0.73% and 0.48%, respectively. Given that the C_v of V_{th_pos} , V_{hold_pos} , V_{th_neg} , and V_{hold_neg} extracted from the highly uniform steady state I-V curves in Fig. 2b were 0.65%, 0.86%, 0.31% and 1.68%, respectively, we conclude that such devices exhibit very low threshold voltage variations even when it is connected to external circuits.

To address this question, we have included the results in Supplementary Fig. 4 and a brief discussion in the main text of the revised manuscript:

- Page 8-9:

“Fig. 2c displays the cumulative plots of positive and negative threshold/holding voltages, including V_{th_pos} , V_{hold_pos} , V_{th_neg} , and V_{hold_neg} in 100 repeated cycles. The coefficient of variation (C_v) defined by the ratio of the standard deviation (σ) to the mean value (μ) of V_{th_pos} , V_{hold_pos} , V_{th_neg} , and V_{hold_neg} were 0.65%, 0.86%, 0.31% and 1.68%, respectively, showing very low cycle-to-cycle (C2C) variations. The superior uniformity can be attributed to the high crystallinity epitaxial VO₂ thanks to the matching lattice planes across the film-substrate interface⁴⁸, as well as the preservation of such desirable qualities in a planar device structure (Supplementary Note 1). In addition to the uniformity observed under steady state, the VO₂ memristor also

displayed very small variations in V_{th} and V_{hold} when it was connected to an external circuit and was operating in a dynamical state (Supplementary Fig. 4). The C_v of V_{th} and V_{hold} during ~ 1000 periods of transient oscillations were 0.73% and 0.48%, respectively.”

Supplementary Figure 4. V_{th} and V_{hold} variations under dynamical conditions. (a)

The extraction of V_{th} and V_{hold} of a VO₂ memristor when it was connected to an external neuron circuit and undergoing oscillations. **(b)** The distribution of the extracted V_{th} and V_{hold} from ~ 1000 periods of oscillation illustrating very low variations.

4. How reliable is the architecture in encoding the ECG/EEG signals by using a VO₂ as an active material?

Our response: We would like to thank the reviewer for raising this important question.

The reliability of the VO₂ memristor-based encoding architecture can be evaluated from two angles: the endurance of the VO₂ device to ensure the durability of the encoder and the low cycle-to-cycle (C2C) variation in V_{th} to ensure accurate encoding of the analog input.

We characterized the endurance of the VO₂ memristor by connecting it in a neuron circuit and repeatedly subjecting it to pulses with a width of 500 μs. Each period of oscillation consists of a pair of transitions from HRS to LRS and then back to HRS, which counts as a switching cycle. As evident in Supplementary Fig. 6, the VO₂ memristor can withstand $>6.5 \times 10^6$ switching cycles without any noticeable degradation in the oscillating behavior and the *I-V* characteristics, showcasing very high endurance.

We performed further simulations to evaluate the effect of V_{th} fluctuations on the encoding quality of the proposed encoder. To emulate V_{th} fluctuations, we superimposed Gaussian noise with a zero mean and varying standard deviations ($\sigma = 0.00$ V, 0.01 V, 0.02 V, 0.03 V, 0.05 V, 0.08 V, 0.10 V, 0.13 V and 0.15 V) on the constant V_{th} in our VO₂ SPICE model. At a V_{th} of 3.4 V, this is equivalent to C_v values ranging from 0% to 4.41%. We randomly selected 8 ECG signals and simulated the noisy encoding process ten times for every combination of signal and σ , totaling 720 trials. In each trial, different random noise was regenerated. The mean squared error (MSE) between the original and the reconstructed signal was calculated for each trial. The average MSE for each C_v and the individual MSE of each trial are plotted in Supplementary Fig. 17a. Also indicated in the figure is the C_v of V_{th_pos} of our VO₂ device. In general, the MSE increases with C_v as a larger fluctuation degrades encoding quality. The distribution of the individual MSE also increases with C_v . Examples of the reconstructed signal, voltage across the VO₂ device and V_{th} for zero, moderate ($C_v = 2.35\%$) and high ($C_v = 4.41\%$) degrees of V_{th} fluctuation are plotted in Supplementary Fig. 17b-d, respectively. By inspecting the signal reconstruction in Supplementary Fig. 17c and comparing its MSE with the individual MSEs for $C_v < 0.88\%$ ($\sigma < 0.03$ V), we can infer that the proposed encoder based on our VO₂ memristor can perform accurate

spike encoding. The tight MSE distribution at low C_v will also contribute to achieving superior repeatability in spike encoding.

Therefore, these results prove that the encoder employing VO₂ as an active material is a very reliable architecture.

To address this question, we have added Supplementary Fig. 6, Supplementary Fig. 17 and Supplementary Note 2 in the revised manuscript, along with further discussions in the main text:

- Page 9:

“Crucially, the VO₂ memristor demonstrated a high endurance of $>6.5 \times 10^6$ switching cycles (Supplementary Fig 6), which ensures the reliability of encoders and neurons that incorporate such devices.”

- Page 19-20:

“A key aspect that needs to be considered when using a VO₂ memristor in the asynchronous spike encoder is its reliability in encoding physiological signals. This can be assessed based on the lifespan of the encoder and the signal encoding quality. As aforementioned, the VO₂ memristor has a high endurance (Supplementary Fig. 6), which will ensure the durability of the encoder. On the other hand, the quality of signal encoding is affected by V_{th} fluctuations. We introduced varying degrees of V_{th} fluctuations in the SPICE model and performed multiple noisy encoding processes (Supplementary Note 2). The encoding quality was quantified by the mean squared error (MSE) between the original and the reconstructed signals. The results are plotted in Supplementary Fig. 17, along with examples of signal reconstruction under zero, moderate and high degrees of V_{th} fluctuations. As the MSE and C_v correlate positively,

our VO₂ memristor, which has a remarkably low C_v, will yield accurately encoded spike outputs (Supplementary Note 2). Moreover, the tight MSE distribution at such low C_v will also enable superior repeatability in spike encoding. Therefore, these results on the endurance and signal encoding quality attest to the reliability of our VO₂ memristor-based spike encoding architecture.”

- Supplementary Note 2: The impact of V_{th} fluctuations on spike encoding quality

“To emulate V_{th} fluctuations, we superimposed Gaussian noise with a zero mean and varying standard deviations ($\sigma = 0.00\text{ V}, 0.01\text{ V}, 0.02\text{ V}, 0.03\text{ V}, 0.05\text{ V}, 0.08\text{ V}, 0.10\text{ V}, 0.13\text{ V}$ and 0.15 V) on the constant V_{th} in our VO₂ SPICE model. At a V_{th} of 3.4 V, this is equivalent to C_v values ranging from 0% to 4.41%. We randomly selected 8 ECG signals and simulated the noisy encoding process ten times for every combination of signal and σ , totaling 720 trials. In each trial, different random noise was regenerated. The mean squared error (MSE) between the original and the reconstructed signal was calculated for each trial.

The average MSE for each C_v and the individual MSE of each trial are plotted in Supplementary Fig. 17a. Also indicated in the figure is the C_v of V_{th_pos} of our VO₂ device. In general, the MSE increases with C_v as a larger fluctuation degrades encoding quality. The distribution of the individual MSE also increases with C_v. Examples of the reconstructed signal, voltage across the VO₂ device and V_{th} for zero, moderate (C_v = 2.35%) and high (C_v = 4.41%) degrees of V_{th} fluctuation are plotted in Supplementary Fig. 17b-d, respectively. By inspecting the signal reconstruction in Supplementary Fig. 17c and comparing its MSE with the individual MSEs for C_v < 0.88% ($\sigma < 0.03\text{ V}$), we can infer that the proposed encoder based on our VO₂ memristor can perform accurate

spike encoding. The tight MSE distribution at low C_v will also contribute to achieving superior repeatability in spike encoding.

Supplementary Figure 6. Endurance of the VO₂ memristor. (a) Stable oscillating behavior of the VO₂ memristor circuit after 10^2 , 10^3 , 10^4 , 10^5 , 10^6 and 6.5×10^6 switching cycles. The endurance was tested by connecting the VO₂ memristor in a neuron circuit and repeatedly subjecting it to pulses with a width of 500 μ s. **(b)** Stable I - V characteristics after 10^2 , 10^3 , 10^4 , 10^5 , 10^6 and 6.5×10^6 switching cycles. **(c)** V_{th} and V_{hold} extracted from the I - V characteristics.

Supplementary Figure 17. The influence of C_v of V_{th} on the quality of signal encoding. (a) A scatter plot of the individual MSE between the original and the reconstructed signals for each of the 720 encoding trials and the average MSE for each C_v . The dotted line indicates the C_v of V_{th_pos} of our VO₂ memristor. (b)-(d) The original ECG signal (upper panel, blue), reconstructed signal (upper panel, red), voltage across the VO₂ memristor (middle and lower panel, green), and V_{th} (middle and lower panel, orange) under (b) zero, (c) moderate and (d) high degrees of fluctuations.

5. The memristor behavior of VO₂ as well as the physiological signal processing using other CMOS devices and memristor are also known. Hence the author

should clearly highlight the advantage of VO₂ based memristor for physiological signal processing in clear terms. The mentioned aspect of efficiency etc are not very clear and appeared to be very generic.

Our response: We would like to thank the reviewer for pointing out the ambiguity regarding the mentioned efficiency of our proposed system in the main text.

The proposed VO₂ memristor-based physiological signal processing system has a high area efficiency. To illustrate this, we compared each VO₂ memristor-based module with various other CMOS and memristor implementations.

First, we estimated the area of the VO₂ memristor-based neurons by considering the size of each individual circuit component. After appropriate circuit-level optimizations (Supplementary Table 10), our LIF and ALIF neurons occupy areas of only ~41.3 μm^2 and ~53.4 μm^2 , respectively. These figures are three to four orders of magnitude better than those before optimizations. Our ALIF neuron also shows at least ~2.2 \times area gains when compared to other implementations (Supplementary Table 11). One of the optimizations is to substantially reduce the capacitors as they dominate the area of the neurons. This is possible owing to the ability of VO₂ in switching at ultrafast timescales¹¹. Besides, the large resistors in the ALIF neuron should be replaced by transistors operating as current source loads. For the sake of completeness, the speed of the ALIF neurons in terms of the spiking frequency as well as the energy consumption are also given in Supplementary Table 11 (see also Supplementary Fig. 29), wherein we considered an optimized VO₂ memristor with a lower V_{th} and a higher HRS based on ref⁷. We then performed a more comprehensive comparison between ALIF implementations based on a figure of merit (FOM) given by

$$FOM = \frac{f}{E \cdot A}$$

where f , E and A are the spiking frequency, energy per spike and area, respectively. As summarized in Supplementary Table 11, our VO₂ memristor-based ALIF neuron can also achieve the highest FOM after optimizations.

Next, we estimated the area of the VO₂ memristor-based asynchronous spike encoder. In this case, the area is dominated by the op-amps and the capacitors. Here, an optimized VO₂ memristor was again considered⁷. Taking into account the capability of the op-amp in driving the VO₂ load circuit, the area of the op-amp was assumed to be 725 μm^2 as reported in ref¹². Furthermore, the capacitors were also reduced while maintaining the encoding functionality of the architecture. Along with the size of other individual circuit components (Supplementary Table 12), the area was estimated to be $\sim 2231 \mu\text{m}^2$. As summarized in Supplementary Table 13, the optimized VO₂ memristor-based encoder is almost an order of magnitude more compact than other similar encoders.

Therefore, we conclude that with appropriate device and circuit optimizations, a physiological processing system that employs our VO₂ memristor-based encoder and neurons is superior to other CMOS or memristor implementations primarily in terms of area benefits.

We would like to further elaborate on the future of densely integrated VO₂ memristor-based physiological signal processing architecture in next-generation human-machine interfaces. Further shrinking of VO₂ memristors is desirable in realizing hardware-based neural networks with an even higher integration density, especially in neuron circuits when capacitors, which are the dominant area-consuming components, are reduced or even replaced by the intrinsic parasitic capacitance for faster computations. Planar devices with gap sizes of 100 nm or less have been reported

previously^{6,7}, and aggressive scaling down to the limits of lithography is possible given that the metal-insulator transition and, subsequently, the threshold switching behavior still exists at the nanoscale^{8,9}. Apart from illustrating the benefits of our proposed physiological signal processing system, another takeaway from the above discussion on the efficiency of our system is the need for meticulous co-optimizations between various circuit components. The demonstrated co-optimizations, although simple, represent the first of many steps that need to be emphasized. Lastly, we further envision the merging of the VO₂ memristor-based encoders and neurons with non-volatile crossbar arrays of emerging memories¹⁰ via proper interfacing (Supplementary Fig. 30) to ultimately realize an extremely compact physiological signal processing architecture.

To address this issue, we have added Supplementary Fig. 29, 30 and Supplementary Tables 10-13, along with the following discussions in the main text and Supplementary Note 3 of the revised manuscript:

- Page 32-33:

“The proposed VO₂ memristor-based physiological signal processing system has a high area efficiency. To illustrate this, we compared each VO₂ circuit module with existing CMOS or memristor implementations (Supplementary Note 3). With proper device and circuit optimizations (Supplementary Table 10), the LIF and ALIF neuron can achieve a small area of ~41.3 μm^2 and ~53.4 μm^2 , respectively. Besides achieving the smallest area overhead, it is worth noting that the optimized VO₂ memristor-based ALIF neuron is also superior in terms of the combined aspects of area, speed and energy consumption (Supplementary Fig. 29, Supplementary Table 11). Furthermore, the proposed VO₂ memristor-based encoder can achieve an area of ~2231 μm^2 , which is almost an order of magnitude smaller than other similar encoders (Supplementary

Table 12-13). Thus, VO₂ memristor-based encoder and neurons can provide substantial benefits over other CMOS or memristor implementations in realizing physiological signal processing systems. Further shrinking of VO₂ memristors is desirable in realizing hardware-based neural networks with an even higher integration density, especially in neuron circuits when capacitors, which are the dominant area-consuming components, are reduced or even replaced by the intrinsic parasitic capacitance for faster computations. Planar devices with gap sizes of 100 nm or less have been reported previously^{40,61}, and aggressive scaling down to the limits of lithography is possible given that the metal-insulator transition and, subsequently, the threshold switching behavior still exists at the nanoscale^{62,63}. Apart from illustrating the benefits of our proposed physiological signal processing system, another takeaway from this discussion is the need for meticulous co-optimizations between various circuit components. The demonstrated co-optimizations, although simple, represent the first of many steps that need to be emphasized. Lastly, we further envision the merging of the VO₂ memristor-based encoders and neurons with non-volatile crossbar arrays of emerging memories²⁷ via proper interfacing (Supplementary Fig. 30) to ultimately realize an extremely compact physiological signal processing architecture.”

- Supplementary Note 3: A comparison between different implementations of neurons and asynchronous spike encoders

“The proposed VO₂ memristor-based physiological signal processing system has a high area efficiency. To illustrate this, we compared each VO₂ memristor-based module with various other CMOS and memristor implementations.

First, we estimated the area of the VO₂ memristor-based neurons by considering the size of each individual circuit component. After appropriate circuit-level

optimizations (Supplementary Table 10), our LIF and ALIF neurons occupy areas of only $\sim 41.3 \text{ um}^2$ and $\sim 53.4 \text{ um}^2$, respectively. These figures are three to four orders of magnitude better than those before optimizations. Our ALIF neuron also shows at least $\sim 2.2\times$ area gains when compared to other implementations (Supplementary Table 11). One of the optimizations is to substantially reduce the capacitors as they dominate the area of the neurons. This is possible owing to the ability of VO_2 in switching at ultrafast timescales²⁰. Besides, the large resistors in the ALIF neuron should be replaced by transistors operating as current source loads. For the sake of completeness, the speed of the ALIF neurons in terms of the spiking frequency as well as the energy consumption are also given in Supplementary Table 11 (see also Supplementary Fig. 29), wherein we considered an optimized VO_2 memristor with a lower V_{th} and a higher HRS based on ref³. We then performed a more comprehensive comparison between ALIF implementations based on a figure of merit (FOM) given by

$$FOM = \frac{f}{E \cdot A}$$

where f , E and A are the spiking frequency, energy per spike and area, respectively. As summarized in Supplementary Table 11, our VO_2 memristor-based ALIF neuron can also achieve the highest FOM after optimizations.

Next, we estimated the area of the VO_2 memristor-based asynchronous spike encoder. In this case, the area is dominated by the op-amps and the capacitors. Here, an optimized VO_2 memristor was again considered¹³. Taking into account the capability of the op-amp in driving the VO_2 load circuit, the area of the op-amp was assumed to be 725 um^2 as reported in ref⁴. Furthermore, the capacitors were also reduced while maintaining the encoding functionality of the architecture. Along with the size of other individual circuit components (Supplementary Table 12), the area was estimated to be

$\sim 2231 \text{ um}^2$. As summarized in Supplementary Table 13, the optimized VO_2 memristor-based encoder is almost an order of magnitude more compact than other similar encoders.

Therefore, we conclude that with appropriate device and circuit optimizations, a physiological processing system that employs our VO_2 memristor-based encoder and neurons is superior to other CMOS or memristor implementations primarily in terms of area benefits.”

Supplementary Figure 29. Energy consumption of the optimized LIF and ALIF neurons. (a) Energy per spike of the optimized LIF neuron obtained by integrating the transient power ($I_{in} \times V_m$) over time and then dividing the result by the number of spikes. **(b)** Energy per spike of the optimized ALIF neuron obtained using the same procedure, with the transient power being the sum of $I_{in} \times V_m$ and the power drawn by the adaptive control circuit ($I_{vdd} \times V_{dd}$).

Supplementary Figure 30. An overview of a compact hardware-based spiking neural network. To achieve high-density integration, such a network may employ our VO₂ memristor-based asynchronous spike encoder to convert analog signals into spikes, as well as VO₂ memristor-based LIF and ALIF neurons to perform biologically plausible neural computations. Further area savings can be made by utilizing dense non-volatile crossbar arrays of emerging memories for synaptic computations.

Supplementary Table 10. Parameters of the optimized ALIF

V_{dd}	1.2 V
I_{in}	26 μ A
R_1	4 k Ω
R_2 (PMOS current source as load)	$L = 260$ nm, $W = 65$ nm, $V_{g,bias} = 0.6$ V
R_3 (NMOS current source as load)	$L = 1040$ nm, $W = 65$ nm, $V_{g,bias} = 0.4$ V
C_1	100 fF
C_2	30 fF
M_1	$L = 65$ nm, $W = 65$ nm
M_2	$L = 100$ nm, $W = 100$ nm
M_3	$L = 65$ nm, $W = 65$ nm

Supplementary Table 11. Detailed comparison between adaptive LIF circuits

	Area (μm^2)	Energy per spike (J)	Adapted frequency (Hz)	FOM (Hz/ $(\mu\text{m}^2 \cdot \text{J})$)	Notes
Indiveri et al. (2006) ¹	2573	900×10^{-12}	-	-	• CMOS-based
Indiveri (2007) ²	-	-	-	-	• CMOS-based
Ferreira et al. (2019) ³	120	3.6×10^{-15}	205×10^3	4.75×10^{17}	• CMOS-based
Ahmadi-Farsani et al. (2022) ⁴	863	140×10^{-12}	1×10^3	8.28×10^9	• CMOS-based
Wang et al. (2018) ⁵	~ 320	-	30×10^6	-	• Memristor-based • Area estimated by calculating capacitor area only and using a capacitance density of 2.5 fF/ μm^2 .
Shaban et al. (2021) ⁶	11435.7	158×10^{-12}	-	-	• Memristor-based • Energy based on reported values for TSMC 65 nm ASIC simulation and spike count for SMNIST.

This work	~224000	6.92×10^{-9}	100×10^3	6.45×10^7	 • Memristor-based • Unoptimized. • Area estimated using a capacitance density of $2.5 \text{ fF}/\mu\text{m}^2$.
	~53.4	338×10^{-15}	33.2×10^6	1.84×10^{18}	 • Memristor-based • Optimized. • VO_2 based on ref¹³. • Circuit based on PTM 65 nm MOSFETs. • Area estimated using a capacitance density of $2.5 \text{ fF}/\mu\text{m}^2$ and a sheet resistance of $690 \Omega/\square$.

Supplementary Table 12. Parameters of the optimized spike encoder

Electric source	$V_{\text{dd}} = 1.2 \text{ V}, V_{\text{ss}} = -1.2 \text{ V}$
M_1	$W = 23 \mu\text{m}, L = 2.5 \mu\text{m}$
M_2	$W = 13 \mu\text{m}, L = 2.5 \mu\text{m}$
D_1 (Diode-connected PMOS)	$W = 1.3 \mu\text{m}, L = 65 \text{ nm}$
D_2 (Diode-connected NMOS)	$W = 0.7 \mu\text{m}, L = 65 \text{ nm}$
C_1	800 fF
C_2	800 fF
R_1	500 Ω
R_2	16.5 k Ω
R_3	1 k Ω
R_4	100 k Ω
R_5	100 k Ω

Supplementary Table 13. Detailed comparison between spike encoders

	Area (μm^2)	Notes
Sharifshazileh et al. (2021) ⁷	~13000	 • CMOS-based • Area estimated from chip design layout provided
He et al. (2021) ⁸	~17618	 • CMOS-based

		 • Area estimated from chip photograph provided
This work	~2231	 • Memristor-based • Optimized. • VO₂ based on ref¹³. • Op-amp based on ref¹⁴. • Circuit based on PTM 65 nm MOSFETs. • Area estimated using a capacitance density of 2.5 fF/μm² and a sheet resistance of 690 Ω/□.

6. Although the application part is presented well, the scientific findings or mechanism part is not standing out. It will be good, if the author describes the mechanism of memristor with more details. For example, the mentioned phase transition induced filament formation do not have any direct evidence and needs to be strengthened.

Our response: We would like to thank the reviewer for the valuable suggestion.

To strengthen the discussion on the switching mechanism of the VO₂ memristor, we have performed a thermodynamic simulation of the device using COMSOL Multiphysics (Supplementary Fig. 3) and included further discussions on the Joule heating-induced phase transition and filament formation in the main text:

- Page 8:

“This resistive switching phenomenon arises from the metal-insulator transition of VO₂, which is a result of the intertwined structural and electronic phase changes⁴²⁻⁴⁴. The transition between the low-temperature semiconducting phase and the high-temperature metallic phase occurs at around ~340 K, and can be triggered by Joule heating⁴⁵. To illustrate this point, we simulated the thermodynamic resistive switching

process using COMSOL Multiphysics. As shown in Supplementary Fig. 3, the switching of the VO₂ memristor between HRS and LRS is accompanied by the formation or disappearance of a high-temperature filament, which has also been previously observed^{46,47}. To be specific, ...”

and

“The simulated I-V curve agrees well with the experimentally measured curve, further verifying the Joule heating-induced phase transition and the filament formation picture.”

Supplementary Figure 3. COMSOL simulation results. (a) The simulated I-V curve agrees well with the experimental I-V curve. (b)-(c) Surface temperature maps of the VO₂ channel corresponding to points (1) and (2) on the I-V curve in (a), respectively. The switching of the device between HRS and LRS is accompanied by the formation or disappearance of a high-temperature metallic filament.

References

- R1 Jerry, M., *et al.* Stochastic Insulator-to-Metal Phase Transition-Based True Random Number Generator. *IEEE Electron Device Lett.* **39**, 139-142 (2018).
- R2 Qazilbash, M. M., *et al.* Mott Transition in VO₂ Revealed by Infrared Spectroscopy and Nano-Imaging. *Science* **318**, 1750-1753 (2007).
- R3 Narayan, J. & Bhosle, V. M. Phase transition and critical issues in structure-property correlations of vanadium oxide. *J. Appl. Phys.* **100**, 103524 (2006).
- R4 Yi, W., *et al.* Biological plausibility and stochasticity in scalable VO₂ active memristor neurons. *Nat. Commun.* **9**, 4661 (2018).
- R5 Nandi, S. K., *et al.* Understanding modes of negative differential resistance in amorphous and polycrystalline vanadium oxides. *J. Appl. Phys.* **128**, 244103 (2020).
- R6 Aetukuri, N. P. B., Harris, J. S., McIntyre, P. C. & Parkin, S. S. P. The control of metal-insulator transition in vanadium dioxide. Stanford University, (Department of Materials Science and Engineering, 2013).
- R7 Dutta, S., *et al.* Programmable coupled oscillators for synchronized locomotion. *Nat. Commun.* **10**, 3299 (2019).
- R8 Bohaichuk, S. M., *et al.* Fast Spiking of a Mott VO₂-Carbon Nanotube Composite Device. *Nano Lett.* **19**, 6751-6755 (2019).
- R9 Bohaichuk, S. M., *et al.* Localized Triggering of the Insulator-Metal Transition in VO₂ Using a Single Carbon Nanotube. *ACS Nano* **13**, 11070-11077 (2019).
- R10 Duan, Q., *et al.* Spiking neurons with spatiotemporal dynamics and gain modulation for monolithically integrated memristive neural networks. *Nat. Commun.* **11**, 3399 (2020).

- R11 Frougier, J., *et al.* Phase-Transition-FET exhibiting steep switching slope of 8mV/decade and 36% enhanced ON current. In: *2016 IEEE Symposium on VLSI Technology* (2016).
- R12 Yosefi, G. The high recycling folded cascode (HRFC): A general enhancement of the recycling folded cascode operational amplifier. *Microelectron. J.* **89**, 70-90 (2019).

REVIEWERS' COMMENTS

Reviewer #1 (Remarks to the Author):

The answers provided by the authors answered my concerns, not only by providing complementary applications, but also by providing new data that clarifies my points. I don't have any other concern, I think the manuscript is suitable for Nature Communications.

Reviewer #2 (Remarks to the Author):

The article by Yang et al titled "A neuromorphic physiological signal processing system based on VO₂ memristor for next-generation human-machine interface" is an interesting advancement in the field of functional oxides and suitable for publication. I have gone through the response letter and the author addressed all my comments satisfactorily and I recommend the revised manuscript for the consideration of acceptance in Nature Communication.

While the author addressed the comments in detailed manner, I hope the length of paper and focus will be ensured in this detailed work by moving few things to supplementary information suitably.

MS No: NCOMMS-23-03137A

Title: A neuromorphic physiological signal processing system based on VO₂ memristor for next-generation human-machine interface

Response to the editor and the reviewers

We would like to sincerely thank the editor for the kind consideration of our manuscript. We are delighted to find that both reviewers have recommended the publication of the manuscript. In light of the minor suggestion from reviewer #2, we have made small adjustments to the manuscript.

Reviewer #1 (Remarks to the Author):

The answers provided by the authors answered my concerns, not only by providing complementary applications, but also by providing new data that clarifies my points. I don't have any other concern, I think the manuscript is suitable for Nature Communications.

Our response: We are very pleased that the reviewer finds our manuscript suitable for publication and would like to once again thank the reviewer for the positive remark.

Reviewer #2 (Remarks to the Author):

The article by Yang et al titled "A neuromorphic physiological signal processing system based on VO₂ memristor for next-generation human-machine interface" is an interesting advancement in the field of functional oxides and suitable for

publication. I have gone through the response letter and the author addressed all my comments satisfactorily and I recommend the revised manuscript for the consideration of acceptance in Nature Communication.

Our response: We are delighted that the reviewer recommends the acceptance of our manuscript and would like to also thank the reviewer for the positive comment.

While the author addressed the comments in detailed manner, I hope the length of paper and focus will be ensured in this detailed work by moving few things to supplementary information suitably.

Our response: We would like to thank the reviewer for the constructive suggestion. As such, we have moved the derivation of the mathematical model for the spiking frequency of the LIF neuron from Pages 9-10 to Supplementary Note 2 to ensure the conciseness of the main text.

- Page 9:

“These firing behaviors can also be deduced from the RC circuit analysis detailed in Supplementary Note 2.”

- Supplementary Note 2: An RC circuit analysis of the LIF neuron

“The artificial LIF neuron is essentially an RC circuit. Starting from the voltage differential equation, the time evolution of the capacitor voltage can be derived and the charging time t_r from V_{hold} to V_{th} can then be expressed by Eq. 1:

$$t_r = [R_L \parallel (R_{off} + R_0)] C_p \cdot \ln \left(\frac{\frac{R_{off} + R_0}{R_L + R_{off} + R_0} V_{in} - V_{hold}}{\frac{R_{off} + R_0}{R_L + R_{off} + R_0} V_{in} - V_{th}} \right) \quad (1)$$

R_{on} and R_{off} denote the resistance of the LRS and HRS, respectively. Similarly, the discharging time t_f from V_{th} to V_{hold} can be described by Eq. 2:

$$t_f = [R_L \parallel (R_{on} + R_0)] C_p \cdot \ln \left(\frac{V_{th} - \frac{R_{on} + R_0}{R_L + R_{on} + R_0} V_{in}}{V_{hold} - \frac{R_{on} + R_0}{R_L + R_{on} + R_0} V_{in}} \right) \quad (2)$$

Thus, the frequency can be calculated by Eq. 3:

$$f = \frac{1}{t_r + t_f} \quad (3)$$

As t_r dominates the spiking period, the frequency is then approximately equal to the inverse of t_r . Subsequently, a brief analysis of the effect of circuit parameters on the frequency can be made. Observe that the natural logarithmic term is a decreasing function of the inner voltage dividing term, which in turn increases with V_{in} and decreases with R_L . Besides, the term $R_L \parallel (R_{off} + R_0)$ decreases as R_L decreases. Thus, a smaller R_L or a larger V_{in} decreases t_r and increases the frequency. Furthermore, t_r is a linear function of C_p , hence a smaller C_p also decreases t_r and increases the frequency.”